

# Characterisation of the artificial neural network CiPS for cirrus cloud remote sensing with MSG/SEVIRI

Johan Strandgren[1], Jennifer Fricker[1], and Luca Bugliaro[1]

[1]Deutsches Zentrum für Luft- und Raumfahrt, Institut für Physik der Atmosphäre, Oberpfaffenhofen, Germany

*Correspondence to:* Johan Strandgren (johan.strandgren@dlr.de)

**Abstract.** Cirrus clouds remain one of the key uncertainties in atmospheric research. To better understand the properties and physical processes of cirrus clouds, accurate large scale observations from satellites are required. Artificial neural networks (ANNs) have proved to be a useful tool for cirrus cloud remote sensing. Since physics is not implemented explicitly in ANNs, a thorough characterisation of the networks is necessary.

In this paper the CiPS (Cirrus Properties from SEVIRI) algorithm is characterised using the space-borne lidar CALIOP. CiPS is composed of a set of ANNs for the cirrus cloud detection, opacity identification and the corresponding cloud top height, ice optical thickness and ice water path retrieval from the imager SEVIRI aboard the geostationary Meteosat Second Generation satellites. First, the retrieval accuracy is characterised with respect to different land surface types. The retrieval works best over water as well as vegetated surfaces, whereas a surface covered by permanent snow & ice or barren reduces the cirrus detection

ability and increases the retrieval errors for the ice optical thickness and ice water path if the cirrus cloud is thin. Second, the retrieval accuracy is characterised with respect to the vertical arrangement of liquid, ice clouds and aerosol layers as derived from CALIOP lidar data. The CiPS retrievals show little interference from liquid water clouds and aerosol layers below an observed cirrus cloud. Only for thin cirrus clouds with an optical thickness below 0.3 or ice water path below $5.0\,\mathrm{g\,m^{-2}}$, a liquid water cloud vertically close or adjacent to the cirrus clearly increases the average retrieval errors for the optical thickness and ice water path respectively. For the cloud top height retrieval, only aerosol layers affect the retrieval error, with an increased

positive bias when the cirrus is at low altitudes. Third, the CiPS retrieval error is characterised with respect to the properties of the investigated cirrus cloud (ice optical thickness and cloud top height). On average CiPS can retrieve the cirrus clouds top height with a relative error around 8 % and no bias and the ice optical thickness with a relative error around 50 % and bias around $\pm 10$ % for the most common combinations of cloud top height and ice optical thickness. Similarities with physically

based retrieval methods are evident, which implies that even though the retrieval methods differ in the physical implementation, the retrievals behave similarly due to physical constraints. Finally, we also show that the ANN retrievals have a low sensitivity to radiometric noise in the SEVIRI observations. For optical thickness and ice water path the relative uncertainty due to noise is less than 10 % down to sub-visual cirrus. For the cloud top height retrieval the uncertainty due to noise is around 100 m for all cloud top heights.



# 1   Introduction

Cirrus clouds remain one of the key uncertainties in atmospheric research (e.g. Waliser et al., 2009; Eliasson et al., 2011; Stevens and Bony, 2013). In particular, the net radiative forcing of cirrus clouds strongly depends on optical thickness, which mainly determines top-of-atmosphere (TOA) reflected shortwave radiation, and cloud height, which together with optical thickness defines the outgoing longwave flux (Meerkötter et al., 1999). To monitor and understand the properties of cirrus clouds, accurate and quantitative large scale observations from satellites are required. Different satellite sensors used for cloud remote sensing have their individual strengths and weaknesses. Imaging radiometers like SEVIRI (Spinning Enhanced Visible InfraRed Imager, Schmetz et al., 2002), ABI (Advanced Baseline Imager, Schmit et al., 2015), MODIS (Moderate Resolution Imaging Spectroradiometer, King et al., 1992) and AVHRR (Advanced Very High Resolution Radiometer, Hastings and Emery, 1992), measure the longwave radiation emitted by the Earth and the reflected solar radiation leaving the Earth-Atmosphere system at TOA. Imaging radiometers possess a large spatial coverage required to observe complete cloud systems, but lack the vertical component and have a limited sensitivity to thin and sub-visual (visible optical thickness < 0.03) cirrus clouds. Active sensors like CALIOP (Cloud-Aerosol Lidar with Orthogonal Polarization, Winker et al., 2003, 2009) and CPR (Cloud Profiling Radar, Stephens et al., 2002) emit visible (CALIOP) and microwave radiation (CPR) and measure the radiation backscattered by clouds and aerosols. This allows for vertical profiling of clouds and aerosols along the satellite track with a high sensitivity to thin cirrus clouds (using the lidar), but leads to a poor spatial coverage between orbits.

Observations from different satellite orbits generate additional advantages and limitations. Sensors observing the Earth from polar orbits (e.g. MODIS, AVHRR, CALIOP and CPR) have a near-global coverage and high spatial resolution, but a low to poor temporal resolution depending on the swath width/spatial coverage. In contrast, a geostationary imager like SEVIRI lacks a global coverage, but has a constant large field of view, which allows for a high temporal resolution of 15 minutes (Schmetz et al., 2002) required to study the temporal evolution, life cycle and physical processes of clouds.

The advantages of individual instruments can be combined to enhance cloud retrievals if two or more complementary satellite sensors operate aboard the same satellite platform (e.g. the synergistic retrievals for the IIR thermal camera and CALIOP aboard CALIPSO by Garnier et al., 2012, 2013, 2015) or fly in a satellite constellation like the A-train (e.g. the synergistic retrievals for CALIOP and CPR or CALIOP, CPR and MODIS by Donovan and van Lammeren, 2001; Deng et al., 2010; Ceccaldi et al., 2013; Delanoë and Hogan, 2008, 2010).

Combining the advantages from satellite sensors operating in different orbits is more challenging, as they observe given scenes at different times from possibly different perspectives. Artificial neural networks (ANNs) have proven to be a powerful tool for this (Kox et al., 2014; Holl et al., 2014; Minnis et al., 2016; Strandgren et al., 2017). With ANNs the relationship between observations of one set of sensors and the retrieval outcome of another set of sensors can be approximated using available sensor collocations. Kox et al. (2014) developed an ANN-based algorithm trained with coincident SEVIRI thermal observations and CALIOP products for the cloud top height (CTH) and ice optical thickness (IOT) determination of cirrus clouds from SEVIRI. Strandgren et al. (2017) exploit the main idea of Kox et al. (2014) and combine four ANNs trained with SEVIRI thermal observations, model data and CALIOP products for the detection of thin cirrus clouds and the retrieval of



the corresponding CTH, IOT and ice water path (IWP) along with an additional opacity information. Holl et al. (2014) utilise ANNs trained with coincident CALIOP, CPR, AVHRR and MHS (Microwave Humidity Sounder) retrievals for the detection and IWP determination of ice clouds from AVHRR and MHS observations. Minnis et al. (2016) estimate the optical thickness of opaque ice clouds at night using an ANN trained with collocated MODIS IR observations and CPR retrievals. The ultimate goal

with these approaches is to retrieve CALIOP (CALIOP/CPR, CPR resp.) like cloud properties from SEVIRI (AVHRR/MHS, MODIS resp.) observations alone. Although ANNs are a powerful alternative to physically based cloud retrievals (e.g. Platnick et al., 2003; Bugliaro et al., 2011; Minnis et al., 2011; Stengel et al., 2014; Heidinger et al., 2015; Wang et al., 2016; Iwabuchi et al., 2016), they are trained to learn patterns and model relationships, and physical principles are not imposed for the scenes being investigated. Consequently, it is difficult to predict how sensitive the retrievals are to different land surface types and

the wealth of natural atmospheric situations and if there are physical conditions where the ANNs are incapable of retrieving meaningful results. This might be due to a number of reasons. On one side the channels of the imager possess a limited vertical resolution expressed by the channel weighting functions as a result of the wavelength and temperature dependent absorption, emission and scattering interactions with gas, cloud and aerosol layers in the atmosphere. On the other hand ANNs will in general perform better for retrieval scenes that occur more frequently in the training dataset as those scenes will have a stronger

weight during the training. This, together with the fact that ANNs provide no direct uncertainty estimates, highlights the importance of properly characterising the ANN retrievals.

In this paper we address those aspects by characterising the CiPS (Cirrus Properties from SEVIRI, Strandgren et al., 2017) algorithm in order to increase the understanding about the functionality, performance and robustness of the corresponding ANNs under different, sometimes challenging, retrieval conditions. SEVIRI and CALIOP, the main input and output training

data sources used for CiPS, represent very different instruments. The active CALIOP lidar provides vertical profiles of ice cloud extinction and makes use of polarisation to distinguish between liquid water and ice. The CiPS algorithm finally exploits the brightness temperature of the Earth sensed by the passive imager SEVIRI to detect ice clouds and derive their optical and physical properties. Although the physics underlying the two retrievals have similarities, like the fact that both instruments saturate at relatively low IOTs (between 3 and 5), the measurement types are very different. Thus, one shall evaluate in detail how

effective the combination of these two instruments is in reality and if there are situations where their different characteristics lead to unreliable results.

In Sect. 2 the satellite sensors and data used used for the characterisation are briefly introduced. The CiPS algorithm is described in Sect. 3. The relative weight/importance of the input variables used by CiPS is estimated in Sect. 4.2. This information is valuable for further ANN developments within the field of (cirrus) cloud remote sensing. In Sect. 4.3 the CiPS retrieval

accuracy is characterised as a function of the underlying surface type, using a set of five surface type classes extracted from MODIS L3 data. In Sect. 4.4 we compare the retrieval accuracy of CiPS for scenes with clear air, aerosol layers as well as high and low liquid water clouds below the cirrus, using vertical characteristics of aerosol and cloud layers derived from CALIOP L2 data. Furthermore we analyse the retrieval errors of CiPS as a function of IOT and CTH in Sect. 4.5, in order to quantify the retrieval error for different types of cirrus. This provides valuable information about the retrieval errors that is not obtained by

looking at the errors averaged across all CTH and/or IOT. In Sect. 4.6 the noise sensitivity of CiPS is quantified by comparing





**Table 1.** Radiometric noise of MSG-2/SEVIRI thermal channels (first column, including the channel centre wavelength $\lambda_c$) at the reported reference brightness temperatures (second column, EUMETSAT, 2007) and at typical brightness temperatures observed for cirrus cloud retrievals (third column, see Sect. 4.6.1)

| Channel, $\lambda_c$ | NE$\Delta$T / K | |
|---|---|---|
| 6.2 μm | 0.05 @ 250 K | 0.11 @ 225 K |
| 7.3 μm | 0.05 @ 250 K | 0.07 @ 237 K |
| 8.7 μm | 0.075 @ 300 K | 0.15 @ 252 K |
| 10.8 μm | 0.07 @ 300 K | 0.12 @ 253 K |
| 12.0 μm | 0.10 @ 300 K | 0.16 @ 251 K |
| 13.4 μm | 0.205 @ 270 K | 0.27 @ 239 K |

the standard retrieval of CiPS with randomly perturbed SEVIRI input data. Finally the results are summarised and discussed in the concluding section. A list of abbreviations is available in Appendix A.

## 2 Instruments and data

### 2.1 SEVIRI

The SEVIRI imager operates aboard the geostationary Meteosat Second Generation (MSG) satellites. SEVIRI measures the up-welling radiation within 12 wavelength intervals (channels) in the visible to thermal infrared spectrum, from which the radiances, equivalent black body brightness temperatures and reflectances can be derived. SEVIRI has a spatial coverage from approx. 80° W to 80° E and 80° S to 80° N (from now on referred to as the *SEVIRI disc*) and a temporal resolution of 15 minutes. Limiting the spatial coverage to latitudes north of approx. 15° N, the temporal resolution can be increased to 5 minutes using

the rapid scanning service. The spatial sampling of SEVIRI is 3 km at nadir for all channels except the high resolution visible (HRV) channel that has a spatial sampling of 1 km (Schmetz et al., 2002).

The radiometric noise levels of the SEVIRI thermal channels are reported as noise equivalent temperature differences (NE$\Delta$T) at given reference temperatures in EUMETSAT (2007) and summarised in Table 1 (second column) for all channels (first column) used by CiPS (see Sect. 3.2). As the reference temperatures reported by EUMETSAT are higher than typical

brightness temperatures observed by SEVIRI for cirrus covered pixels, the third column shows the radiometric noise scaled to reference brightness temperatures representing typical cirrus cloud retrievals (see Sect. 4.6.1).

### 2.2 CALIOP

The CALIOP lidar observes the Earth from a polar orbit aboard CALIPSO (Cloud-Aerosol Lidar and Infrared Pathfinder Satellite Observations). CALIOP emits 20 laser pulses per second and measures curtains of attenuated backscatter profiles

along the satellite track with a vertical resolution of up to 30 m (Winker et al., 2009). In this study we use the V3 CALIOP L2



cloud and aerosol layer data at a spatial resolution of 5 km (CAL_LID_L2_05kmC|ALay-Prov-V3-0X..., CALIPSO Science Team, 2016a, b). The layer products provide the vertical position of cloud and aerosol layers in the atmosphere, as well as cloud phase, optical thickness, ice water path and opacity. The opacity is reported as a binary flag and tells whether CALIOP was able to fully penetrate the layer (transparent) or not (opaque).

## 2.3 Collocation dataset

For this study (Sect. 4.3, 4.4, 4.5, 4.6) we use a dataset of collocated CiPS input data (SEVIRI, ECMWF and auxiliary data, see Sect. 3.2) and cirrus properties retrieved by CALIOP (CTH, IOT, IWP and opacity information), allowing us to apply CiPS and compare the retrievals with the corresponding reference retrievals by CALIOP. The cirrus properties retrieved by CALIOP have been collocated with the SEVIRI observations from the pixel having the largest overlap with the 5 km CALIOP orbit segment. Due to the different viewing geometries of SEVIRI and CALIOP, the latitude, longitude and cloud top altitude from CALIOP were used to project cirrus clouds to the SEVIRI grid (parallax correction). The dataset contains 5 million collocations collected over a time period of almost 6 years (April 2007 to January 2013) and was initially used to validate CiPS. A detailed description of the dataset can be found in Strandgren et al. (2017).

## 3 The CiPS algorithm

To improve the readability of the paper, artificial neural networks and the CiPS algorithm are shortly introduced.

### 3.1 Artificial neural networks

An artificial neural network (ANN) is a mathematical model that can be trained to recognise patterns and model functions. A set of multilayer perceptrons (MLPs), a feed-forward artificial neural network, is used by CiPS for the remote sensing of cirrus clouds and is thus shortly introduced here. The goal of an ANN is to model the relationship between two sets of data, such that a vector of output data can be accurately estimated using the information from a vector of known input data. A MLP consists of a number of neurons that exchange information with each other. The neurons are distributed over three major units; 1) the input layer that holds as many neurons as input variables, 2) the output layer that holds as many neurons as output variables and 3) the hidden layers that hold an arbitrary number of hidden neurons distributed over a number of hidden layers. The output value of a neuron is calculated by processing the output from all neurons in the preceding layer connected to that neuron and the corresponding numeric weights assigned to each neuron-neuron connection through an activation function. Hence the only information available to solve a problem is the input data and all connection weights. Thus it is crucial that the weights are assigned correct values.

The weights are tuned by training the ANN. Using the backpropagation algorithm (Rumelhart et al., 1986), as is the case for CiPS, the weights are tuned by looking at a large number of training examples, where both the input data and the corresponding output data are known. The ANN uses the training input data and the current weights to calculate a vector of output data. The skill of the ANN is determined by calculating the squared error between the vector of estimated output data and the



corresponding vector of known reference output data. The squared error is then propagated backwards through the ANN and each weight is tuned such that the error is minimised. The training procedure is an iterative process and continues until the error between estimation and reference is sufficiently low. For more details, see Strandgren et al. (2017).

## 3.2 CiPS

The Cirrus Properties from SEVIRI (CiPS) algorithm (details in  Strandgren et al., 2017) detects cirrus clouds, identifies opaque pixels and retrieves the corresponding CTH, IOT and IWP. To this end a set of four ANNs are used, trained with MSG-2/SEVIRI thermal observations, the surface skin temperature $T_{\mathrm{surf}}$ (from ECMWF) and auxiliary data as input and V3 CALIOP L2 layer data (CALIPSO Science Team, 2016a, b) as reference output data. CiPS uses one ANN to derive a cirrus cloud flag (CCF) that classifies the SEVIRI pixels as either cirrus free or cirrus covered. A second ANN is used for the CTH

retrieval and a third ANN for the IOT and IWP retrieval. The fourth ANN is used to derive an opacity flag (OPF) that classifies the cirrus covered pixels as either transparent or opaque. As CALIOP becomes saturated for optically thicker cirrus (IOT $\gtrsim 3$), the CiPS IOT and IWP retrievals should not be trusted in such situations. Thus the OPF is trained to distinguish between the cirrus clouds that could be fully penetrated by CALIOP (transparent cirrus) and those that could not (opaque cirrus).

   CiPS input data selection is based on physical considerations. CiPS works pixel by pixel and uses the single brightness

temperatures from the SEVIRI channels centred at 6.2, 7.3, 8.7, 10.8, 12.0 and 13.4 μm. Water vapour channels (centred at 6.2 and 7.3 μm) should help detecting ice clouds (see e.g.  Krebs et al., 2007), identifying opaque pixels as well as determining its height, together with the $CO_2$ channel centred at 13.4 μm (e.g.  Menzel et al., 1983; Schmetz et al., 1993). Window channels (8.7, 10.8, 12.0 μm) and especially their brightness temperature differences are both useful for detection (e.g. Inoue, 1985) and for the optical thickness determination (e.g. Ackerman et al., 1990). Furthermore CiPS exploits the information from nearby

SEVIRI pixels by utilising the regional maximum brightness temperature from the window channels (for all ANNs, as a proxy for cirrus free conditions) and the regional average brightness temperature from the water vapour channels (only for cirrus detection and opacity classification, as a proxy for the smoothness of the surroundings). The regional maximum brightness temperature is defined as the maximum brightness temperature within a $19 \times 19$ pixels large box (corresponding to an area of $\approx 57 \times 57 \, \mathrm{km}^2$ at nadir) centred at the pixel under consideration. Similarly the regional average brightness temperature

is defined as the boxcar average temperature within the same box (inspired by Krebs et al., 2007). The modelled surface temperature from ECMWF provides a cirrus-free characterisation of the surface and should be useful in all ANNs. Finally, CiPS uses the latitude, the viewing zenith angle of SEVIRI, two surface type flags (sea water and permanent ice and snow) and the day of the year (DOY, 1–365: to avoid a hard transition from December 31 to January 1, two input neurons are used for the DOY: $\sin(2\pi \, \mathrm{DOY}/365)$ and $\cos(2\pi \, \mathrm{DOY}/365)$). Latitude and day of year are selected since the appearance of cirrus and

their top height strongly depends on general circulation and convection strength, with higher clouds in the tropics and generally lower clouds towards the polar regions, and with stronger convection in Summer with respect to Spring/Autumn and, of course, Winter in mid-latitudes. Viewing angle shall account for the path length of radiation through the atmosphere, while the two selected surface types identify on one side (sea) thermally quite homogeneous surfaces and on the other side (ice/snow) cold surfaces with similar absorption properties as the ice clouds. In total, 18 input variables are used for the cirrus detection and



opacity classification and 16 input variables for the CTH, IOT and IWP retrieval. Although the selection of input quantities is inspired by physical principles, the task of combining input variables is left to the ANN.

In the following, all quantities referring to CiPS will be denoted as $CCF_{CiPS}$, $OPF_{CiPS}$, $CTH_{CiPS}$, $IOT_{CiPS}$ and $IWP_{CiPS}$, while all quantities referring to CALIOP will be denoted as $CTH_{CALIOP}$, $IOT_{CALIOP}$ and $IWP_{CALIOP}$

## 4 Characterisation of CiPS

Strandgren et al. (2017) presents the CiPS retrieval accuracy for cirrus detection, opacity classification and for the derivation of the physical and optical properties CTH, IOT and IWP with respect to CALIOP. In this paper, a more differentiated investigation is performed that aims at characterising the ANNs according to various aspects. First, despite the fact that CiPS input quantities have been selected according to physical principles (See Sect. 3.2), it is unclear which importance the single input variables have been assigned by the ANNs. This a posteriori examination also gives hints about the ability of the ANNs to model physical relationships among the variables. Second, the combination of cirrus products from visible backscattered vertically resolved "monochromatic" lidar radiation (CALIOP) and thermal "columnar" narrowband brightness temperatures from imager channels (SEVIRI) is supported by the knowledge that cirrus clouds leave their mark on both measurements types in a "similar" way: for instance, both methods are sensitive to visible ice optical thickness up to 5 ca. (e.g. DeSlover et al., 1999). Nevertheless, CALIOP's possibility of discerning vertical features (ice clouds, liquid water clouds, aerosols) is not shared by SEVIRI, which poses the question whether the proposed CALIOP-SEVIRI synergy is always meaningful. To clarify this aspect, the CiPS performance is investigated for various vertical arrangements of cloud and aerosol layers and for various surface types. Furthermore, cirrus clouds are classified according to their IOT and CTH to provide a better understanding of the CiPS retrieval errors (magnitude and bias). Finally the sensitivity to radiometric noise in the SEVIRI data is quantified.

### 4.1 Validation metrics

First of all, the validation metrics used in the following are presented.

The probability of detection (POD) is used to measure how efficiently CiPS detects cirrus clouds and is given by

$$\text{POD} = \frac{N_{TP}}{N_{TP} + N_{FN}} \, , \tag{1}$$

where the number of true positives, $N_{TP}$, are all points correctly classified as cirrus and the number of false negatives, $N_{FN}$, all cirrus clouds that remain undetected. The denominator, $N_{TP} + N_{FN}$, is thus the total number of points with a reference cirrus cloud. The false alarm rate (FAR) measures the fraction of cirrus free points that are falsely classified as being cirrus clouds and is given by

$$\text{FAR} = \frac{N_{FP}}{N_{FP} + N_{TN}} \, , \tag{2}$$

where the number of false positives, $N_{FP}$, are all points falsely classified as cirrus (false alarms) and the number of true negatives, $N_{TN}$, all points correctly identified as cirrus free. The denominator, $N_{FP} + N_{TN}$, is thus the total number of points





**Table 2.** Contingency table for the cirrus detection from CALIOP and CiPS.

|  |  | CALIOP | |
|---|---|---|---|
|  |  | Cirrus | No cirrus |
| CiPS | Cirrus | $N_{TP}$ | $N_{FP}$ |
|  | No cirrus | $N_{FN}$ | $N_{TN}$ |

with no reference cirrus cloud. The corresponding CALIOP data are used as a reference when calculating the POD and FAR. Table 2 clarifies the quantities used to calculate the POD and FAR.

The mean absolute percentage error (MAPE) and mean percentage error (MPE) are defined as

$$\text{MAPE} = \frac{100\%}{N} \sum_{i=1}^{N} \left| \frac{E_i - O_i}{O_i} \right|, \tag{3a}$$

$$\text{MPE} = \frac{100\%}{N} \sum_{i=1}^{N} \frac{E_i - O_i}{O_i}, \tag{3b}$$

where $O_i$ is the observed reference value retrieved by CALIOP and $E_i$ the estimated value by CiPS. The sum spans over all samples $i = 1, \ldots, N$ used for the evaluation. The MAPE gives information about the average magnitude of the CiPS retrieval errors relative to the expected reference value retrieved by CALIOP. The MPE gives information about the direction of the deviations, i.e. whether CiPS tends to overestimate (positive MPE) or underestimate (negative MPE) the values with respect to CALIOP (bias). When calculating the MPE, over- and underestimates can cancel out each other, potentially leading to zero MPE/bias even if the magnitude of the errors is large. Therefore the MAPE has been considered as well.

## 4.2 Relative weight of the CiPS input data

To understand, improve and extend CiPS and similar ANN-based retrieval algorithms, it is valuable to understand what input data have essential contributions to the solution of a given problem. Important input variables are identified by the ANN and given a strong weight during the training. Similarly, less important input variables are given a weaker weight and thus a smaller role in retrieving the output data.

The weight of an input variable can be estimated as the euclidean length of the vector holding all weights that connect that input neuron with the hidden neurons in the first hidden layer (LeCun et al., 1990). Figure 1 shows the relative weight of the 18 input variables used by CiPS (the sum of the relative weights across all input variables adds up to 100 % for each ANN). The four lines represent the four ANNs. For the CTH$_{\text{CiPS}}$, IOT$_{\text{CiPS}}$ and IWP$_{\text{CiPS}}$ retrievals, no relative weight is reported for the regional average brightness temperatures since those are used exclusively for the cirrus detection and opacity classification (see Sect. 3.2).

It is clear that the window channels of SEVIRI are essential for the detection and opacity classification of cirrus clouds as well as for the determination of IOT$_{\text{CiPS}}$ and IWP$_{\text{CiPS}}$. This reflects the importance of these channels in physically based retrievals





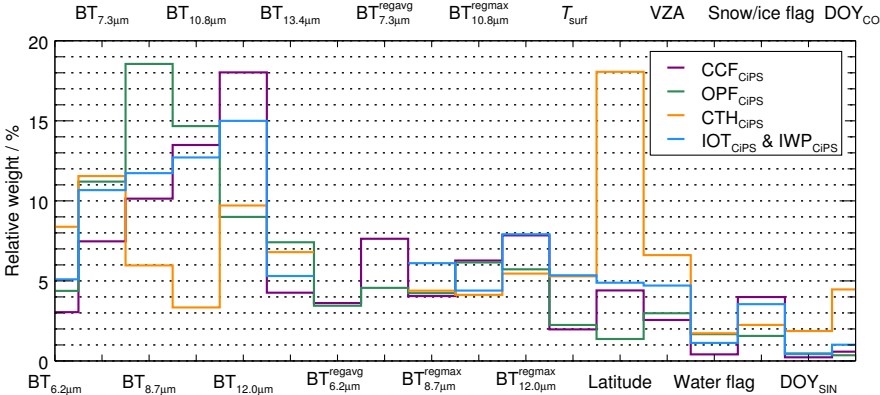

**Figure 1.** Relative weight of the CiPS input variables for the 4 ANNs. BT = Brightness Temperature, regavg = regional average, regmax = regional maximum, VZA = viewing zenith angle, $DOY_{SIN} = \sin(2\pi\,DOY/365)$ and $DOY_{COS} = \cos(2\pi\,DOY/365)$.

(e.g. Ewald et al., 2013; Heidinger et al., 2015; Iwabuchi et al., 2016). For the $CTH_{CiPS}$ retrieval, the latitude is the dominant input variable, followed by the water vapour channels. Similarly the DOY has a comparably strong weight for the $CTH_{CiPS}$ retrieval. The weight of $BT_{13.4\mu m}$ is surprisingly low for the $CTH_{CiPS}$ retrieval, although observations from around 13.4 µm are commonly used by the $CO_2$-slicing method for CTH retrievals (e.g. Menzel et al., 2008). This is a hint that the ANN may

model statistical, rather than physical, relationship between the input and output variables, as the CTH has an annual cycle and a clear latitude dependency (Stubenrauch et al., 2013). It might also be that the 13.4 µm brightness temperature only provides redundant information with respect to cloud top height since also water vapour channels and surface skin temperatures are available to the ANN (see discussion about the physical meaning of the input variables in Sect. 3.2). For the $CCF_{CiPS}$, $OPF_{CiPS}$, $IOT_{CiPS}$ and $IWP_{CiPS}$ retrievals, the DOY has a very low weight and consequently a minor contribution to the retrievals. The

surface temperature from the model is clearly helpful for determining the $CTH_{CiPS}$, $IOT_{CiPS}$ and $IWP_{CiPS}$. The information whether the Earth's surface is covered by permanent ice or snow is valuable for the cirrus detection as well as the $IOT_{CiPS}$ and $IWP_{CiPS}$ retrievals, whereas the surface water flag has a comparably small contribution to the retrievals. Exploiting the information from nearby SEVIRI pixels using the regional maximum and regional average temperatures is clearly helpful in all aspects, their weights are comparable to the weight of $T_{surf}$ for the $CTH_{CiPS}$, $IOT_{CiPS}$ and $IWP_{CiPS}$ retrievals for example.

## 4.3 The CiPS retrieval accuracy for different surface types

In this section the performance of CiPS is characterised with respect to a set of five surface type classes extracted from MODIS L3 data. For this section as well as for the remainder of this paper (except the noise sensitivity analysis in Sect. 4.6) the performance of CiPS is always evaluated with respect to the cirrus cloud retrievals by CALIOP.





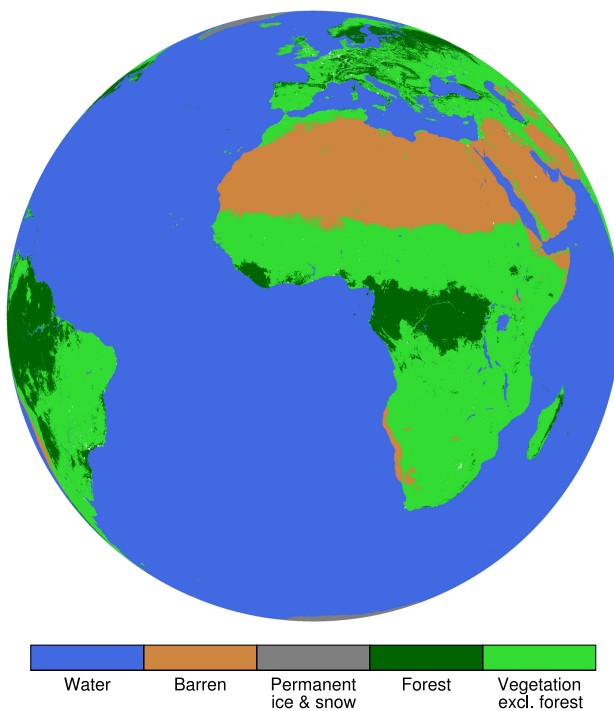

**Figure 2.** Visualisation of the geographical coverage of the five surface type classes across the SEVIRI disc.

### 4.3.1 Surface type classes from MODIS

The International Geosphere-Biosphere Programme (IGBP Loveland and Belward, 1997) has defined 17 land surface types including 11 natural vegetation classes, 3 developed and mosaicked land classes, and 3 non-vegetated land classes. The MODIS L3 product MCD12C1 (Friedl et al., 2010) provides the majority land cover type at a resolution of $0.05°$ according to the IGBP classification. The MCD12C1 dataset for 2012 V051 has first been reprojected to the SEVIRI grid using the nearest neighbour method. Then, for the characterisation of CiPS with respect to the underlying surface type, the different surface classes have been grouped into the five following classes: 1) *Water* including ocean, lakes, rivers and wetlands, 2) *Barren* including surfaces covered by soil, sand and rocks with a maximum vegetation of $10\,\%$, 3) *Permanent ice & snow* including surfaces permanently covered by ice and/or snow, 4) *Forest* including all surfaces dominated by trees (canopy cover $> 60\,\%$) and 5) *Vegetation excl. forest* including all surfaces with other types of vegetation i.e. shrublands, savannahs, grasslands and croplands. Detailed information about the IGBP surface types can be found in Loveland and Belward (1997). The geographical coverage of the five surface classes used in this study are visualised in Fig. 2. These surface types are expected to have different spectral properties and humidity contents that might affect the thermal SEVIRI channels (Sect. 3.2) and therefore the CiPS ANNs.





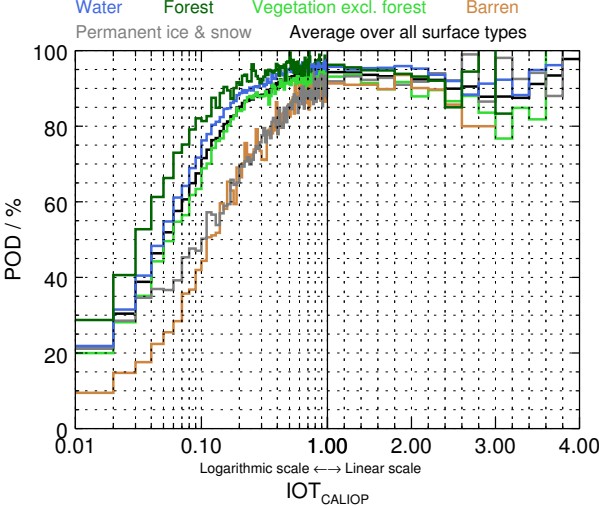

**Figure 3.** The cirrus POD of CiPS as a function of the IOT retrieved by CALIOP for the five surface type classes. Note that the line colours corresponds to the colours used in Fig. 2.

### 4.3.2 Cirrus cloud detection

The CCF$_\text{CiPS}$ (Sect. 3.2) is evaluated as a function of the underlying surface type using the POD (Eq. 1) and the FAR (Eq. 2). To avoid the presence of (liquid water/aerosol) layers between cirrus and surface that would shield radiation emitted by the surface, the CALIOP L2 data are used to identify and include only those profiles with clear air (at most a faint aerosol layer

with aerosol optical thickness AOT $\leq 0.2$) below a possible transparent cirrus cloud in the analysis (see class C1 (for POD) and class C7 (for FAR) in Sect. 4.4.1).

Figure 3 shows the POD of CiPS for the five surface type classes. The POD is presented as a function of IOT$_\text{CALIOP}$. For a better visualisation of thin cirrus, the scale is logarithmic for IOT$_\text{CALIOP} < 1.0$ and linear for IOT$_\text{CALIOP} \geq 1.0$. As a reference, the average POD for all surface types is included (black line). We require a lower limit of 10 samples for the statistics, so no

POD over barren is available for thicker cirrus clouds (IOT$_\text{CALIOP} \gtrsim 3.0$). The same is done for the remainder of this paper.

CiPS has a clearly lower POD over barren and permanent ice & snow for cirrus clouds with IOT$_\text{CALIOP} \lesssim 0.5$: up to 20 % less cirrus clouds are detected than on average. Both are known to be difficult (cirrus) cloud retrieval conditions (Frey et al., 2008; Holz et al., 2008). Over ice and snow the radiative contrast between the cirrus and the cold surface is reduced, making the cirrus cloud detection more difficult. Furthermore, mixed phase clouds or supercooled liquid water layers above ice layers

in the polar regions (Mioche et al., 2015; Verlinde et al., 2007; Shupe et al., 2006) as well as temperature inversions may also reduce the POD as CiPS requires the water to be frozen to be classified as a cirrus. Furthermore, the retrieval conditions over Greenland and Antarctica are the least favourable ones for SEVIRI, with the largest viewing zenith angles and pixel sizes. The FAR over permanent ice & snow is 4.3 %, which is higher than the average FAR of 3.2 % over all surface types. Barren





is to a large extent made up by deserts, where cirrus clouds rarely form. Yet they can be found there when they drift towards mid-latitudes after formation in the ITCZ. The ANN is likely to learn such a pattern of low occurrence frequency and thus miss more thin cirrus in those regions. This is supported by the fact that the FAR is lowest over barren where only 1.1 % of the cirrus free cases are falsely classified as cirrus. The highest POD is observed over forest: up to 15 % more than on average for

$IOT_{CALIOP}$ up to 0.5. This is due to the high cirrus cloud occurrence over the tropical rain forests that increases the POD in a similar manner as the POD is reduced over barren. Again this is supported by the highest FAR of 7.1 % over forest. Similar trends could be seen in Fig. 5 in Strandgren et al. (2017), with the minimum FAR over the Sahara desert and the maximum FAR above the African and South American rain forests. Water and other vegetation (vegetation excl. forest) have similar POD, but the cirrus detection is slightly better over homogeneous water surfaces than over vegetation excl. forest. The corresponding

FARs are 3.1 % over water and 3.5 % over vegetation excl. forest. Notice finally that due to their large number over the SEVIRI disc the water pixels dominate the average curve.

### 4.3.3 Cirrus cloud properties

Figure 4 shows the MAPE and MPE (Eqs. 3) for the (a) $CTH_{CiPS}$, (b) $IOT_{CiPS}$ and (c) $IWP_{CiPS}$ retrievals as functions of the corresponding reference retrievals by CALIOP and the five surface type classes. Within each $CTH_{CALIOP}$, $IOT_{CALIOP}$ and

$IWP_{CALIOP}$ interval in Fig. 4, the MAPE and MPE given by Eq. (3a) and (3b) is calculated. Please note that the results are presented with a logarithmic scale for $IOT_{CALIOP} < 1.0$ and $IWP_{CALIOP} < 10.0\,\mathrm{g\,m^{-2}}$ and with a linear scale for $IOT_{CALIOP} \geq 1.0$ and $IWP_{CALIOP} \geq 10.0\,\mathrm{g\,m^{-2}}$. The average MAPE and MPE (bias) over all surface type classes are included as reference. Again, we only consider profiles with clear air (no liquid water clouds and AOT $\leq 0.2$ below the cirrus cloud).

Mostly the same patterns of the MAPE and MPE are observed as in Strandgren et al. (2017), namely that CiPS tends to

overestimate the CTH for low cirrus/ice clouds and slightly underestimate the CTH for high cirrus. Similarly the IOT and IWP is predominantly over- and underestimated for the lower and upper extreme values respectively.

Overall, the $CTH_{CiPS}$ retrieval is mostly insensitive to the underlying surface type for $CTH_{CALIOP} > 8.0\,\mathrm{km}$. A stronger underestimation of $CTH_{CALIOP}$ (20–40 %) is however observed over permanent ice & snow for high cirrus clouds ($CTH_{CALIOP} > 12\,\mathrm{km}$). Those are cirrus/ice clouds that extend into the stratosphere. We observe a stronger tendency for underestimations for

lower clouds as well over permanent ice & snow. For the lowermost cirrus clouds, the $CTH_{CiPS}$ retrieval is best over permanent ice & snow. This is most likely due to the fact that the average CTH is lowest in the polar regions, making it easier for the ANN to model and estimate the CTH for low cirrus/ice clouds there. On the contrary, barren, forest and vegetation excl. forest do to a large extent cover regions where the CTH is typically higher, making it more difficult to model and estimate the CTH for low cirrus. Over desert, where the air is dry, it is plausible that the signal from the water vapour channels (which were shown

to have strong weight for the $CTH_{CiPS}$ retrieval in Sect. 4.2) peak at lower altitudes in the atmosphere compared to more moist regions, resulting in biases for the CTH retrieval over barren. On average the CTH is estimated with the lowest MAPE and bias over homogeneous water surfaces.

The underlying surface type has a similar effect on the $IOT_{CiPS}$ retrieval as on the $IWP_{CiPS}$ retrieval. This is expected since the $IWP_{CALIOP}$ used to train CiPS is parametrised from the CALIOP extinction coefficients (Heymsfield et al., 2005) from which



$IOT_{CALIOP}$ is directly derived. For $IOT_{CALIOP} > 0.5$ and $IWP_{CALIOP} > 10.0\,\mathrm{g\,m^{-2}}$, the underlying surface type has no effect on the $IOT_{CiPS}$/$IWP_{CiPS}$ retrievals, i.e. already for these low values of $IOT_{CALIOP}$/$IWP_{CALIOP}$ the characteristics of surface radiation are negligible. For thinner cirrus clouds the retrieval errors increase substantially over permanent ice & snow. This should be related to the effects discussed above, namely the reduced radiative contrast of the cirrus above cold snow and ice

and the unfavourable conditions for SEVIRI in the polar regions. $IOT_{CiPS}$/$IWP_{CiPS}$ retrievals over barren are also less certain for thin cirrus clouds. Deserts are characterised by a lower emissivity at $8.7\,\mathrm{\mu m}$ than at $10.8$ or $12.0\,\mathrm{\mu m}$ (e.g. Hulley et al., 2015; De Paepe and Dewitte, 2009; Trigo et al., 2008). It is possible that this induces larger $IOT_{CiPS}$/$IWP_{CiPS}$ retrieval errors because the ANN cannot localise desert regions unambiguously using only latitude and viewing zenith angle. The retrieval errors over vegetation excl. forest are close or identical to the average performance for all $IOT_{CALIOP}$ and $IWP_{CALIOP}$. The lowest $IOT_{CiPS}$

and $IWP_{CiPS}$ retrieval errors are again obtained over homogeneous water surfaces as well as over forest.

## 4.4  The CiPS retrieval accuracy for different vertical cloud-aerosol structures

In this section the performance of CiPS is characterised with respect to a set of seven vertical cloud-aerosol structure classes derived from the V3 CALIOP L2 layer products. This is important in order to understand the accuracy of the retrievals for different scattering and absorbing atmospheres. Although the opacity flag already yields a quality information telling when the

$IOT_{CiPS}$ and $IWP_{CiPS}$ retrievals can be trusted, there is still the chance that the passive instrument SEVIRI is not able to deal with all possible vertical arrangements of clouds and aerosols as the active instrument CALIOP does, since SEVIRI lacks the vertical resolution.

The characterisation is performed for all surface types combined. Although the retrieval accuracy shows a dependency on the underlying surface type (sect. 4.3.1), the effect of liquid water clouds and aerosol layers below the cirrus cloud has a similar

effect on the cirrus cloud retrieval over all underlying surface types (not shown here). The patterns and results obtained for all surface types combined (presented here) are consequently to a large extent representative for the single surface types as well. Due to the large coverage of oceans on the the SEVIRI disc (see Fig. 2), the results presented here are however more representative for retrievals over water.

The goal of the opacity flag retrieved by CiPS is to detect cirrus clouds that are opaque, i.e. where the vertical structure

below the cirrus is unknown for CiPS/CALIOP. Consequently, the opacity flag of CiPS is not characterised for the different vertical cloud-aerosol structures but only the quantities CCF, CTH, IOT and IWP.

### 4.4.1  Vertical cloud-aerosol structures from CALIOP

The collocation dataset presented in Sect. 2.3 and first developed in Strandgren et al. (2017) is extended to characterise the entire atmospheric column observed by CALIOP (and SEVIRI). The column optical thickness and the corresponding top and base

heights for aerosol layers, cirrus clouds and liquid water clouds are derived from the CALIOP cloud and aerosol layer products (see Sect. 2.2). The column aerosol optical thickness (AOT) is read from the 'Column_Optical_Depth_Aerosols_532' product contained in the L2 aerosol layer products. The corresponding top and base heights of the upper and lowermost aerosol layers are read from the 'Layer_Base_Altitude' and 'Layer_Top_Altitude' products. Finally the opacity information is retrieved from





the 'Opacity_Flag'. For clouds, the column optical thickness is reported for liquid water and ice clouds combined. The cloud properties, including IOT, liquid water optical thickness, the corresponding top and base heights and the opacity information, are instead derived using the same approach used to derive the cirrus cloud properties in Strandgren et al. (2017). In particular, one has to take into account the spatial resolution at which cloud and aerosol layers are detected as well as the possible vertical

overlap between layers detected at different spatial resolutions. Notice as well that mixed-phase clouds, i.e. layers where ice and supercooled liquid water coexist are classified as either liquid, ice or unknown phase clouds by CALIOP. The high confidence criteria imposed to the CALIOP cloud phase (see below) shall however constrain the selected cloud and cloud profiles to high confidence liquid and high confidence ice clouds. Nevertheless, especially at high latitudes, an uncertainty remains due to the difficult cloud phase determination (Cesana et al., 2016).

The dataset is carefully quality screened and does only include CALIOP profiles where all feature types and phases were determined with high confidence (QA_flag = 3). Furthermore, the quality assessment from Strandgren et al. (2017) is adopted not only for ice clouds but also for liquid water and, especially, for aerosol layers: only constrained or unconstrained retrievals where the initial lidar ratio remained unchanged during the solution process were included (Strandgren et al., 2017).

Using the vertical position, optical thickness and opacity information of all clouds and aerosol layers, the seven vertical
cloud-aerosol structure classes listed below are created. In this study we are interested in the effect of thicker aerosol layers on the cirrus cloud retrieval by CiPS. Therefore we only acknowledge aerosol layers with an $AOT > 0.2$. These aerosols come mainly from desert dust (Weinzierl et al., 2011; Groß et al., 2015), but also from biomass burning (Rosário et al., 2011; Ten Hoeve et al., 2012) or, sometimes, sea salt (Toth et al., 2013). We assume that $AOT \leq 0.2$ is a good approximation for the AOT of typical aerosol loads. In comparison, the rural aerosol model by Shettle (1989) in the boundary layer, background
aerosol above 2 km, spring-summer conditions and a visibility of 50 km represents an AOT of 0.162.

**C1** Profiles where only transparent cirrus clouds (and possible aerosols with $AOT \leq 0.2$) are observed.

**C2** Profiles where cirrus clouds are observed over an aerosol layer with $AOT > 0.2$.

**C3** Profiles where cirrus clouds are observed above a low opaque liquid water cloud. To ensure that the cirrus is well separated from the water cloud, the vertical distance between the two has to be 4.0 km or more. This class aims to capture scenes
with cirrus clouds over low-level clouds. The threshold of 4 km was chosen such that it is applicable both in the tropical regions as well as at higher latitudes, where the vertical separation between high-level cirrus clouds and low-level clouds is smaller.

**C4** Profiles where cirrus clouds are observed vertically close or adjacent to an opaque liquid water cloud. To ensure that the cirrus is close to the water cloud, the vertical distance between the two has to be 0.5 km or less. This spatial separation
value enables to neglect small cloud gaps due to turbulence, evaporation, sedimentation or wind shear inside clouds. This class aims to capture mainly convective clouds with a cirrus shield/anvil.

**C5** Profiles where only opaque liquid water clouds are observed. No cirrus clouds are present.

**C6** Profiles where only an aerosol layer with $AOT > 0.2$ is observed. No clouds are present.



**C7** Profiles where only clear sky or aerosols with AOT $\leq 0.2$ are observed. No clouds are present.

Please note that all liquid water clouds were opaque for the CALIOP lidar. Hence there is the possibility of having a thicker aerosol layer below the liquid water clouds. The effect of the aerosol layer is however assumed to be negligible due to the use of observations in the IR spectrum where the liquid cloud is also opaque. This vertical cloud-aerosol structure information

is extracted and appended to the corresponding collocations contained in the collocation dataset (Sect. 2.3). For a graphical interpretation of the vertical cloud-aerosol structure classification, all classes are visualised in Fig. 5. The number of samples for each class is also indicated; class C7 with more the 1.7 million samples is the most common situation, C4 with less than 14,000 samples the most seldom.

Some CALIOP profiles do not fit into one of the seven classes. For example if the cirrus cloud is opaque or if the vertical

distance between a cirrus cloud and an underlying liquid water cloud is between 0.5–4.0 km. Furthermore all CALIOP retrievals used for the validation of CiPS in Strandgren et al. (2017) do not necessarily pass the quality screening, since liquid water clouds and aerosols are included as well. In total, 75 % of the CALIOP retrievals contained in the collocation dataset passed the quality screening and could be grouped into one of the seven classes. The remaining 25 % were excluded from the present analysis.

### 4.4.2 Cirrus cloud detection

Figure 6 shows the POD of CiPS for the vertical cloud-aerosol structure classes C1–C4, i.e. those classes defined in Sect. 4.4.1 that contain cirrus clouds. The POD is presented as a function of $IOT_{CALIOP}$. For a better visualisation the scale is again logarithmic for $IOT_{CALIOP} < 1.0$ and linear for $IOT_{CALIOP} \geq 1.0$. As a reference, the average POD for all cirrus clouds in the dataset, also those that did not fit any of the four classes C1–C4 (discussed in Strandgren et al., 2017), is included.

The cirrus cloud detection by CiPS show little interference with different vertical cloud-aerosol structures. It is however considerably easier for CiPS to detect a thin cirrus cloud if a liquid water cloud is present vertically close to the base altitude of the cirrus (C4). Even for sub-visual cirrus the POD is close to 60 % in such situations. If the vertical separation between the cirrus cloud and the liquid water cloud is larger ($\geq 4.0$ km, C3), only a marginal increase in POD with respect to profiles with no liquid water cloud below the cirrus is observed. For thicker cirrus clouds with $IOT_{CALIOP} > 1.0$, the POD is close to 100 %

with a liquid water cloud below the cirrus (C3 and C4), compared to the 95 % for scenes with only a transparent cirrus cloud (C1). An aerosol layer has a small effect on the CiPS cirrus detection in general, but for cirrus clouds with $IOT_{CALIOP} < 0.08$ an aerosol layer appears to attenuate the radiative contrast of the cirrus leading to a slightly lower POD.

97 % of the scenes with clear sky (C7) or thicker aerosol layers (C6) are correctly classified as cirrus free (FAR = 3.2 %). An increased average FAR of 5.5 % is obtained if a liquid water cloud is present (C5). This is a result of CiPS falsely classifying

30 some high liquid water clouds as cirrus clouds. Figure 7 shows the FAR for scenes with liquid water clouds (C5) as a function of the liquid water cloud top temperature along with the relative frequency of occurrence of the different cloud top temperatures. It is clear that the colder (higher up) the liquid water cloud is, the higher is the risk of falsely classifying it as a cirrus cloud. At temperatures below -30°C, the FAR is approx. 35–65 %. The relative frequency of such supercooled liquid water clouds is



however low. Most liquid water clouds have a top temperature between -15°C and +15°C, and thus a clearly lower FAR of less than 5.0 %.

### 4.4.3 Cirrus cloud properties

Figure 8 shows the MAPE and MPE for the (a) $CTH_{CiPS}$, (b) $IOT_{CiPS}$ and (c) $IWP_{CiPS}$ retrievals as functions of the corresponding reference retrievals by CALIOP and the vertical cloud-aerosol structure.

Within each $CTH_{CALIOP}$, $IOT_{CALIOP}$ and $IWP_{CALIOP}$ interval in Fig. 8, the MAPE and MPE given by Eq. (3a) and (3b) is calculated. Again the results are presented with a logarithmic scale for $IOT_{CALIOP} < 1.0$ and $IWP_{CALIOP} < 10.0 \, \mathrm{g m^{-2}}$ and with a linear scale for $IOT_{CALIOP} \geq 1.0$ and $IWP_{CALIOP} \geq 10.0 \, \mathrm{g m^{-2}}$. The average retrieval errors for all vertical cloud-aerosol structures are included as a reference and stem from Strandgren et al. (2017).

The presence of liquid water clouds (C3 and C4) has a negligible effect on the $CTH_{CiPS}$ retrieval. An aerosol layer below the cirrus cloud introduces a stronger positive bias (positive MPE), with a MAPE and MPE of up to 70 % for the lowermost cirrus clouds. This is not necessarily an effect of the aerosol layer itself, and it is likely to be related to the fact that most aerosol layers with AOT > 0.2 are found in the tropical regions (not shown here), where CTHs are typically higher leading to a stronger tendency of overestimating comparably low CTHs. This effect is seen to diminish with increasing $CTH_{CALIOP}$. At $CTH_{CALIOP} = 9.0 \, \mathrm{km}$ the MAPE introduced by an underlying aerosol layer is approx. 5 % larger compared to retrievals without an aerosol layer. Above 13 km, the aerosol layer has no effect on the $CTH_{CiPS}$ retrieval error.

The presence of a low liquid water cloud below the cirrus (C3) has a negligible effect on the $IOT_{CiPS}$ and $IWP_{CiPS}$ retrievals, with the same MPE and MAPE as for situations with solely clear sky or typical aerosols below the cirrus cloud (C1). If the liquid water cloud is located vertically close or adjacent to the cirrus (C4), the retrieval error clearly increases for thin cirrus clouds. The increase in error for those retrievals is seen for $IOT_{CALIOP} \lesssim 0.5$ and $IWP_{CALIOP} \lesssim 10.0 \, \mathrm{g m^{-2}}$ and increases rapidly with decreasing $IOT_{CALIOP}$ and $IWP_{CALIOP}$. At $IOT_{CALIOP} \approx 0.08$ and $IWP_{CALIOP} \approx 2.0 \, \mathrm{g m^{-2}}$, the MAPE is 200 % for class C4, which is about twice the error of the $IOT_{CiPS}$/$IWP_{CiPS}$ retrievals for situations with solely clear sky or typical aerosols below the cirrus cloud (C1). This pattern is to be expected as it is impossible for a radiometer to know where the transition between ice and liquid water occur when the two clouds are not vertically well separated, especially since the liquid water cloud is thick and thus opaque to infrared radiation. Furthermore it is more difficult to extract information from underlying liquid water clouds from the brightness temperature differences, also utilizing the regional maximum brightness temperatures, if the vertical separation and hence the radiative contrast between the cirrus cloud and the underlying liquid water cloud is small. A corresponding increase is observed for the MPE, meaning that the increased MAPE is a result of larger overestimations of $IOT_{CiPS}$ and $IWP_{CiPS}$.

Opposite to the $CTH_{CiPS}$ retrieval, an aerosol layer below the cirrus cloud (C2) reduces the $IOT_{CiPS}$ and $IWP_{CiPS}$ retrieval errors for thin cirrus clouds. This does not imply that it is easier to retrieve the $IOT_{CiPS}$ and $IWP_{CiPS}$ of thin cirrus clouds when an aerosol layer is present below the cirrus. It is rather related to the fact that CiPS predominantly overestimate $IOT_{CALIOP}$ and $IWP_{CALIOP}$ for thin cirrus, an effect that is reduced if an aerosol layer is present below the cirrus.





### 4.5 The CiPS retrieval errors as a function of optical thickness and cloud top height

In this section we investigate the retrieval errors of CiPS as a function of $IOT_{CALIOP}$ and $CTH_{CALIOP}$. This gives information about typical errors of CiPS for different types of cirrus clouds (e.g. low and thick or high and thin cirrus). To remove any effects from different vertical cloud-aerosol structures, again only those profiles with transparent cirrus clouds and possible

faint aerosols (AOT $\leq 0.2$) are used (class C1 in Sect. 4.4.1). In other words, this shows typical CiPS retrieval errors for all transparent cirrus clouds occurring in the collocation dataset (Sect. 4.4.1). This distribution is depicted in Figure 9c, which represents a 2D histogram with $IOT_{CALIOP}$ on the horizontal axis and $CTH_{CALIOP}$ on the vertical axis. The colour map shows the number of occurrences for each combination of $IOT_{CALIOP}$ and $CTH_{CALIOP}$ in the validation dataset. Both the validation dataset and the training dataset used to train CiPS consist of a random subset of CALIOP data collected over a time period

of almost 6 years and do consequently represent the natural distribution of IOT and CTH frequencies and combinations. The occurrences in Fig. 9c are thus to a large extent representative for the corresponding occurrences in the dataset used to train CiPS as well. The highest occurrences of cirrus clouds in Figure 9c are between 9 and 17 km, with tropical cirrus covering the high altitude cirrus fraction and mid- to low-latitude cirrus covering the low altitude cirrus fraction. Low cirrus clouds are thicker than high cirrus, with an occurrence peak for cirrus with $CTH_{CALIOP}$ between 10.5 and 12.5 km and $IOT_{CALIOP}$ between

0.3 and 1.0. In the following we estimate which a priori retrieval error can be expected for this typical natural cirrus cloud distribution.

Figure 9a and 9b show two 2D histograms with the $IOT_{CALIOP}$ and $CTH_{CALIOP}$ on the horizontal and vertical axes respectively. The colour maps show the MAPE and MPE of the $CTH_{CiPS}$ retrievals with respect to the reference CALIOP data (Eqs. 3).

The $CTH_{CiPS}$ retrieval shows a stable performance with a MAPE between 5–15 % for most combinations of top height and optical thickness. To accurately retrieve the CTH (and IOT/IWP) from the satellite, a clear radiative contrast between the cirrus cloud and the Earth's surface or low liquid water clouds is favourable. For optically thin cirrus clouds, radiation from below has a larger contribution to the observed brightness temperatures, which reduces the radiative contrast between the cirrus cloud and the underlying surface. Similarly, the radiative contrast decreases if the cirrus cloud is located further down

in the atmosphere at warmer temperatures, with more water vapour above the cirrus cloud that makes the interpretation of window channel brightness temperatures and brightness temperature differences more difficult. These effects can be seen in the retrieval errors, with generally decreasing MAPE for increasing $CTH_{CALIOP}$ and $IOT_{CALIOP}$. It is clear that the combination of low and optically thin cirrus induce the maximum $CTH_{CiPS}$ retrieval errors (MAPE $\gtrsim 25$ %) while high and optically thick cirrus induce the minimum $CTH_{CiPS}$ retrieval errors (MAPE $\approx 5$ %). The lowest retrieval errors are observed at high altitudes

($CTH_{CALIOP} \in [15, 17]$ km), where the $CTH_{CiPS}$ can be retrieved with a small error also for sub-visual cirrus. Similar features are observed using an optimal estimation method in Iwabuchi et al. (2016). For thin to sub-visual cirrus clouds, CiPS is more likely to overestimate the CTH (positive MPE). With increasing $IOT_{CALIOP}$, the bias weakens and for $IOT_{CALIOP} > 0.05$ and $CTH_{CALIOP} > 8$ km, CiPS is mostly unbiased (MPE $\approx 0$). As already discussed in Strandgren et al. (2017), the extreme high and low $CTH_{CALIOP}$ are primarily under- and overestimated though, irrespective of $IOT_{CALIOP}$.





A correlation between higher MAPE and a low number of occurrences is evident. For the region of low optically thin cirrus, where the MAPE of the $CTH_{CiPS}$ retrieval is highest, there are only few points. This is further clarified in Fig. 9d, showing the MAPE of the $CTH_{CiPS}$ retrieval as a function of the number of occurrences. Each diamond in Fig. 9d represents one pair of grid boxes in Fig. 9a and Fig. 9c (708 pairs of boxes with valid data are represented). It is clear that the high MAPEs rarely

occur and that most $CTH_{CiPS}$ retrievals have comparably low MAPEs. This gives us primarily three pieces of information: 1) The learning of the ANNs is sensitive to the distribution of the training dataset, leading to difficulties to accurately retrieve the cirrus properties for comparably rare situations. An effort was made to balance the training datasets for CiPS by adding duplicates for some rare situations (Sect. 3.4.2 in Strandgren et al., 2017) to increase their weight during the training. This approach does however not introduce any new information that the ANNs can learn from. Nevertheless, not even a perfectly

balanced dataset is likely to result in an ANN that performs equally good for all kinds of cirrus clouds and retrieval conditions, as certain retrieval conditions have physical limitations, as discussed above for low and optically thin cirrus clouds. We also see that CiPS can retrieve the CTH for high sub-visual cirrus clouds with a low MAPE despite a low number occurrences. 2) With comparably few occurrences, the high MAPEs of CiPS have a small effect for the average usage of CiPS, as the MAPE for the comparably common situations is low. 3) Due to their few occurrences, the high MAPEs of CiPS have a low statistical

value such that these values have to be treated with caution.

On average CiPS can retrieve the CTH with a MAPE around 8 % and zero bias (MPE) for the most common combinations of $CTH_{CALIOP}$ and $IOT_{CALIOP}$. Taking the number of occurrences into account, which represents the natural distribution of transparent cirrus clouds observed by CALIOP, 37 % of all $CTH_{CiPS}$ retrievals have a MAPE of 5 % or less. Another 27 % and 16 % of all retrievals have a MAPE between 5–10 % and 10–15 % respectively.

Figure 10 is similar to Fig. 9, but here the $IOT_{CiPS}$ retrieval errors are in focus. Figure 10a–c again show 2D histograms with $IOT_{CALIOP}$ on the horizontal axes and $CTH_{CALIOP}$ on the vertical axes. The colour maps show (a) the MAPE and (b) the MPE of the $IOT_{CiPS}$ retrievals with respect to the CALIOP reference retrievals and (c) the corresponding number of occurrences for the different $IOT_{CALIOP}/CTH_{CALIOP}$ combinations. Figure 10c is consequently a duplicate of Fig. 9c, but is included twice for the reader's convenience. Please note that the retrieval errors are significantly larger for $IOT_{CiPS}$ compared to $CTH_{CiPS}$

and the axes for the MAPE and MPE now range from 0 to 500 % and from -500 to 500 % respectively. The MAPE of the $IOT_{CiPS}$ retrievals as a function of the number of occurrences is shown in Fig. 10d. Similarly to the $CTH_{CiPS}$ retrievals, the $IOT_{CiPS}$ retrieval errors show clear patterns across the $IOT_{CALIOP}$ and $CTH_{CiPS}$ domains. The large retrieval errors for thin cirrus clouds already shown in Strandgren et al. (2017) are evident, but are seen to decrease with increasing $CTH_{CALIOP}$. Above 14 km CiPS can estimate the IOT with a MAPE (Fig. 10a) of 30–120 % down to sub-visual cirrus clouds. Again,

the combination of low ($CTH_{CALIOP} < 8$ km) and optically thin ($IOT_{CALIOP} < 0.1$) cirrus induces the largest $IOT_{CiPS}$ retrieval errors (MAPE $> 150$ %), while high ($CTH_{CALIOP} > 13$ km) and optically thicker ($IOT_{CALIOP} > 0.06$) cirrus induce the smallest retrieval errors (MAPE between 30–80 % and MPE close to zero). Furthermore, there is a band with $IOT_{CALIOP}$ between 0.2 and 0.5 at 4 km height that expands with $CTH_{CALIOP}$ to reach $IOT_{CALIOP}$ between 0.1 and 1.0 at 16 km where the MAPE is smaller than 50 %. The smallest bias (MPE, Fig. 10b) is observed where the MAPE is lowest and increases slightly with decreasing



$CTH_{CALIOP}$. For $IOT_{CALIOP} > 0.3$, the $IOT_{CiPS}$ retrieval has a negative or zero bias on average (MPE between -80–0 %), whereas for $IOT_{CALIOP} < 0.3$ the $IOT_{CiPS}$ retrieval has no or a positive (up to 400 % or more) bias.

Again an evident correlation between low MAPEs and a high number of occurrences is observed (Fig. 10d). And even though high MAPEs of 800 % are possible, the large majority of the $IOT_{CiPS}$ retrievals have MAPEs between 50–150 %. Please also note that a 800 % MAPE observed at $IOT_{CALIOP} = 0.01$ translates into a small absolute error (0.08). Similar optical thickness retrieval errors are shown for the optimal estimation retrieval by Iwabuchi et al. (2016), demonstrating that the large errors are not an artefact of the ANN, but rather due to physical constraints discussed above. There are approx. 250 points/diamonds with less than 200 occurrences and low MAPE ($< 100$ %). Those points represent cirrus clouds with a comparably high optical thickness ($IOT_{CALIOP} \gtrsim 1.5$). In this region CiPS predominantly underestimate $IOT_{CALIOP}$, meaning that the MAPE of a $IOT_{CiPS}$ retrieval is bounded above by 100 %.

On average CiPS can retrieve the IOT with a MAPE around 50 % and bias around $\pm 10$ % for the most common combinations of $CTH_{CALIOP}$ and $IOT_{CALIOP}$. Taking the number of occurrences into account, again representing the natural distribution of transparent cirrus clouds observed by CALIOP, 55 % of all $IOT_{CiPS}$ retrievals have a MAPE of 50 % or less. Another 28 % of the retrievals have a MAPE between 50–100 %, meaning that only 17 % of the retrievals have a MAPE larger than 100 %.

The corresponding results for the $IWP_{CiPS}$ retrieval are similar to those of the $IOT_{CiPS}$ and are therefore not further presented here.

## 4.6 Noise sensitivity analysis of CiPS

In this section the effect of small noisy perturbations in the input data from SEVIRI propagating through the ANNs is quantified. The noise sensitivity analysis is performed for the $CTH_{CiPS}$, $IOT_{CiPS}$ and $IWP_{CiPS}$ retrievals. The collocation dataset described in Sect. 2.3 is used for this purpose in order to have a large temporal and spatial coverage. CiPS classifies 1.3 million points in the collocation dataset as icy, for which the $CTH_{CiPS}$, $IOT_{CiPS}$ and $IWP_{CiPS}$ is retrieved. Along with the standard CiPS retrieval using the observed SEVIRI brightness temperatures, another 100 retrievals for every point are performed where the SEVIRI brightness temperatures are randomly perturbed within the respective radiometric noise ranges.

### 4.6.1 Perturbing the SEVIRI brightness temperatures

The SEVIRI noise equivalent temperature differences (NE$\Delta$T) reported at reference temperatures as given in EUMETSAT (2007) (see the second column of Table 1) are smaller than 0.1 K for SEVIRI window and water vapour channels and smaller than 0.2 K for the $CO_2$ channel. However, these reference temperatures are higher than typical cirrus cloud retrievals. Therefore the reported noise levels are scaled to the respective cirrus cloud brightness temperatures observed by SEVIRI. In a first step the NE$\Delta$T are converted to the noise equivalent radiance differences (NE$\Delta$R) using the derivative of Planck's law (with respect to temperature $T$) at the reported reference temperatures and respective wavelengths (the centre channel wavelength in the first column of Table 1 is used for this purpose). In a second step the NE$\Delta$R are converted back to NE$\Delta$T at the brightness temperature of the corresponding cirrus cloud retrievals (EUMETSAT, 2017, pers. comm.). This results in an individual noise level for all brightness temperatures observed by SEVIRI and used for the standard CiPS retrieval. So with 1.3 million cirrus





cloud retrievals in the collocation dataset and 9 SEVIRI brightness temperatures as input (6 brightness temperatures and 3 regional maximum temperatures, Sect. 3.2), a total of $9 \times 1.3 \cdot 10^6$ individual radiometric noise levels are obtained.

When the radiometric noise in the respective channels is scaled to the observed brightness temperatures, colder cirrus cloud observations get higher radiometric noise levels compared to warmer observations. The third column in Table 1 shows the

radiometric noise levels for the 6 SEVIRI channels used by CiPS at reference brightness temperatures given by typical cirrus cloud observations. Those reference temperatures constitute the average brightness temperatures observed by the respective channels across all CiPS cirrus clouds retrievals in the collocation dataset. It is clear that the noise levels of the cirrus cloud observations are higher compared to the noise levels at the warmer reference brightness temperatures reported by EUMETSAT (2007).

Each of the $9 \times 1.3 \cdot 10^6$ brightness temperature observations in the collocation dataset is associated with a Gaussian distribution with zero mean and standard deviation provided by the $9 \times 1.3 \cdot 10^6$ individual radiometric noise levels produced above. Each Gaussian distribution is finally sampled randomly 100 times yielding $9 \times 1.3 \cdot 10^6 \cdot 100$ uncorrelated noise perturbations across the different SEVIRI input brightness temperatures. Hence, a set of 100 randomly perturbed retrievals is obtained for each cirrus cloud retrieval in the collocation dataset that can be directly compared to the corresponding standard (unperturbed)

retrieval of CiPS.

### 4.6.2 Noise sensitivity of CiPS

The noise sensitivity of the $CTH_{CiPS}$, $IOT_{CiPS}$ and $IWP_{CiPS}$ retrievals is determined by calculating the root-mean-square deviation (RMSD) between the standard retrievals and the corresponding 100 perturbed retrievals for the 1.3 million icy collocations. The RMSD is defined as

$$\text{RMSD} = \sqrt{\frac{1}{100} \sum_{i=1}^{100} (S - P_i)^2} \qquad (4)$$

where $S$ is the standard CiPS retrieval and $P_i$ are the perturbed retrievals ($i = 1, \ldots, 100$). The sum spans over all 100 perturbed retrievals.

Figure 11 shows the RMSD for (a) $CTH_{CiPS}$, (b) $IOT_{CiPS}$ and (c) $IWP_{CiPS}$ as functions of the respective quantities. For $IOT_{CiPS}$ and $IWP_{CiPS}$ only retrievals classified as transparent by CiPS ($OPF_{CiPS} = 0$) are included. This reduces the number of

samples from 1.3 to approx 1 million. Please note that again the results are presented with a logarithmic scale for $IOT_{CALIOP} < 1.0$ and $IWP_{CALIOP} < 10.0 \, \text{gm}^{-2}$ and with a linear scale for $IOT_{CALIOP} \geq 1.0$ and $IWP_{CALIOP} \geq 10.0 \, \text{gm}^{-2}$. The surface type and the vertical cloud-aerosol structures are not taken into account for the noise sensitivity analysis and the reported results represent the average sensitivity to radiometric noise across all retrieval conditions.

The $CTH_{CiPS}$ retrieval is clearly robust with a low sensitivity to noise in the SEVIRI input data. The RMSD is around 100 m

throughout the whole $CTH_{CiPS}$ range.

The $IOT_{CiPS}$ and $IWP_{CiPS}$ retrievals have similar noise sensitivities. The RMSD is less than 10 % of the corresponding $IOT_{CiPS}$/$IWP_{CiPS}$ throughout most of the $IOT_{CiPS}$ and $IWP_{CiPS}$ ranges. Only for sub-visual cirrus the RMSD of CiPS is higher.



For thicker cirrus, the $IOT_{CiPS}$ and $IWP_{CiPS}$ retrievals become more robust to SEVIRI noise as the respective curves flatten towards a constant sensitivity around 0.1 and $3\,gm^{-2}$ for $IOT_{CiPS}$ and $IWP_{CiPS}$ respectively. For thin cirrus clouds, a small change in IOT/IWP induces a comparably large change in the cloud radiative properties. Similarly a small change in the cloud radiative properties has a larger effect on the IOT and IWP for thin cirrus clouds compared to thicker cirrus where the IOT and

IWP is higher. Consequently a small noisy perturbation applied to the SEVIRI input data has a larger impact on the $IOT_{CiPS}$ and $IWP_{CiPS}$ retrievals for thin cirrus clouds, leading to higher relative RMSD for thin cirrus and decreasing relative RMSD for thicker cirrus.

A noise sensitivity of 0.001 at a retrieved optical thickness of 0.01 is low and one may expect noise to have a strong impact on the retrievals for such faint cirrus. The reported radiometric noise of SEVIRI is however very low. Even for cold cirrus

cloud retrievals, the radiometric noise level is between 0.07–0.27 K on average for the 6 SEVIRI channels (see third column in Table 1), which corresponds to 0.3–1.1‰ of the observed average brightness temperatures. Furthermore the noise is assumed to be Gaussian and peaks at zero across all perturbed retrievals and the individual SEVIRI input variables.

In Sect. 4.5 the retrieval error of CiPS is assessed to 5–15 % and 50–150 % for the $CTH_{CiPS}$ and $IOT_{CiPS}/IWP_{CiPS}$ retrievals respectively. In this section the radiometric noise of SEVIRI is shown to have a minor contribution to the retrieval error. Thus

it is clear that the major part of the retrieval error stems from the clearly different characteristics and sensitivities of SEVIRI and CALIOP.

## 5   Conclusions

The CiPS algorithm (Cirrus Properties from SEVIRI, Strandgren et al., 2017) utilises a set of four artificial neural networks for the geostationary remote sensing of cirrus clouds with MSG/SEVIRI. In Strandgren et al. (2017) the retrieval accuracy was

evaluated over all underlying surfaces types, vertical cloud-aerosol structures and IOT-CTH combinations combined. In this paper we perform a thorough characterisation of CiPS with respect to several aspects in order to 1) learn more about the CiPS retrieval accuracy under various retrieval conditions; 2) learn more about the ANN method for (cirrus) cloud remote sensing; 3) learn more about potential and limitations of the synergistic use of the two, in many aspects, very different instruments, CALIOP and SEVIRI.

Over vegetated surfaces, CiPS retrieves the CTH, IOT and IWP with similar retrieval errors as over homogeneous water (ocean, lakes, rivers and wetlands). Over permanent snow & ice and barren (mostly desert), surface types that are known to induce difficult (cirrus) cloud remote sensing conditions (Frey et al., 2008; Holz et al., 2008), the $IOT_{CiPS}/IWP_{CiPS}$ retrieval errors clearly increases only for thin cirrus clouds ($IOT_{CALIOP} \lesssim 0.3$, $IWP_{CALIOP} \lesssim 5.0\,gm^{-2}$) with respect to retrievals over water. Liquid water clouds below the observed cirrus have only a small or no effect on the $CTH_{CiPS}$ retrieval with respect to

cirrus retrievals with no interfering cloud or aerosol layers below the cirrus. The $IOT_{CiPS}/IWP_{CiPS}$ retrieval errors are clearly increased only if: 1) the cirrus cloud is thin and 2) the liquid water cloud is vertically close or adjacent to the cirrus cloud. A liquid water cloud well separated ($> 4.0\,km$) from the cirrus cloud has no or small effect on the $IOT_{CiPS}/IWP_{CiPS}$ retrieval errors, even for sub-visual cirrus clouds. This shows that the limited vertical resolution of the thermal channels is sufficient





to separate the contributions from these objects. This clearly differentiates CiPS from solar channel based Nakajima-King (Nakajima and King, 1990) retrievals (e.g., Platnick et al., 2003; Bugliaro et al., 2011; Stengel et al., 2014) that are only able to provide the optical thickness of the entire atmospheric column if no additional a priori information is used. The retrieval errors are further shown to decrease with increasing top height and optical thickness. For the most common combinations of

$CTH_{CALIOP}$ and $IOT_{CALIOP}$, CiPS can retrieve the CTH with a MAPE around of 8 % and no bias (MPE = 0) and the IOT with a MAPE of around 50 % and a bias of around $\pm 10$ % on average. Above 14 km, $IOT_{CiPS}$ is retrieved with a MAPE between 30–120 % down to sub-visual cirrus. Similar to physically based retrieval methods (e.g. Iwabuchi et al., 2016), CiPS struggles for low and optically thin cirrus clouds, where the radiative contrast between the cirrus clouds and the underlying surface is weaker. This implies that although CiPS (and ANNs in general) lacks the explicit implementation of physical principles, the

CiPS cirrus cloud retrievals have similar limitations as physically based retrievals where the physical principles are explicitly implemented. We have shown that there are few conditions where CiPS can not retrieve meaningful results. And although significant retrieval errors are possible, we show that most of those conditions are rarely observed from SEVIRI (e.g. cirrus clouds over permanent snow & ice (see Fig. 2), transparent cirrus clouds (as seen by CiPS/CALIOP) vertically close or adjacent liquid water clouds (see Fig. 5) and optically thin *and* low cirrus clouds (see Fig. 9c/10c).

The main limiting factors for the retrievals are shown to be physical constraints induced by for example low and optically thin cirrus clouds that have a weak radiative contrast to underlying liquid water clouds or land surfaces as seen from space. Only a small fraction of the retrieval errors ($\approx 10$ %) is shown to stem from the radiometric noise of SEVIRI. So even though the physical principles are not implemented explicitly in CiPS, the characterisation shows that the ANNs to a large extent could model the physical relationships between the input and output data. There are situations where limiting factors of the

ANN method become apparent as well. For example CiPS underestimates the cirrus presence over desert due to the naturally low probability of cirrus cloud occurrence. Similarly, it overestimates the cirrus presence over tropical rainforests due to the naturally high probability of cirrus cloud occurrence. Likewise, CiPS is more likely to underestimate high $CTH_{CALIOP}$ in regions where cirrus clouds are typically found at low altitudes. We also see that latitude has a considerably stronger weight than the brightness temperature at 13.4 μm for CTH, indicating that CiPS in that case rather modelled a statistical, rather than

physical, relationship between input and output data.

With this paper we have shown that the CiPS retrievals to a large extent show little interference with the underlying land surface type, the vertical cloud-aerosol structure below the cirrus and provide accurate results for common combinations of IOT and CTH. The idea of combining SEVIRI brightness temperature observations at a spatial resolution of 9 km$^2$ or more with vertically resolved CALIOP lidar point measurements averaged over 5 km via a set of ANNs turns out to be very successful,

despite the different measurement principles and sensitivities. There are some conditions where increased retrieval errors are clearly observed though. This proves the importance of such characterisations in order to more efficiently identify limitations and better understand the retrievals under different retrieval conditions. As we see that retrieval limitations to a large extent stem from physical constraints, this conclusions is not restricted to retrievals utilising ANNs.



## Appendix A:  List of abbreviations

ANN       Artificial Neural Network

CCF       Cirrus Cloud Flag

CTH       Cloud Top Height

FAR       False Alarm Rate

IOT       Ice Optical Thickness

IWP       Ice Water Path

MAPE      Mean Absolute Percentage Error

MLP       MultiLayer Perceptron

MPE       Mean Percentage Error

OPF       Opacity Flag

POD       Probability of Detection

RMSD      Root-Mean-Square Deviation

*Acknowledgements.*  This research was supported by the DLR (Deutsches Zentrum für Luft- und Raumfahrt)/DAAD (Deutscher Akademischer Austauschdienst) Research Fellowship Programme für Doktoranden, 14.

5    We thank the NASA Atmospheric Science Data Center for their kind support and for providing the V3 CALIOP layer data in a subsetted form. We are grateful to André Butz for his valuable input on the sensitivity analysis and general feedback on the manuscript. We also acknowledge the constructive comments of Florian Ewald that improved the quality of this manuscript. We gratefully acknowledge Pamela Schöbel-Pattiselanno for her guidance on how to interpret the reported radiometric noise of SEVIRI.

The SEVIRI data were provided by EUMETSAT and the modelled surface temperature was obtained from the European Centre For
10   Medium-Range Weather Forecasts (ECMWF).

The MCD12C1 data product was retrieved from the online Data Pool, courtesy of the NASA Land Processes Distributed Active Archive Center (LP DAAC), USGS/Earth Resources Observation and Science (EROS) Center, Sioux Falls, South Dakota, https://lpdaac.usgs.gov/data_access/data_pool.





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





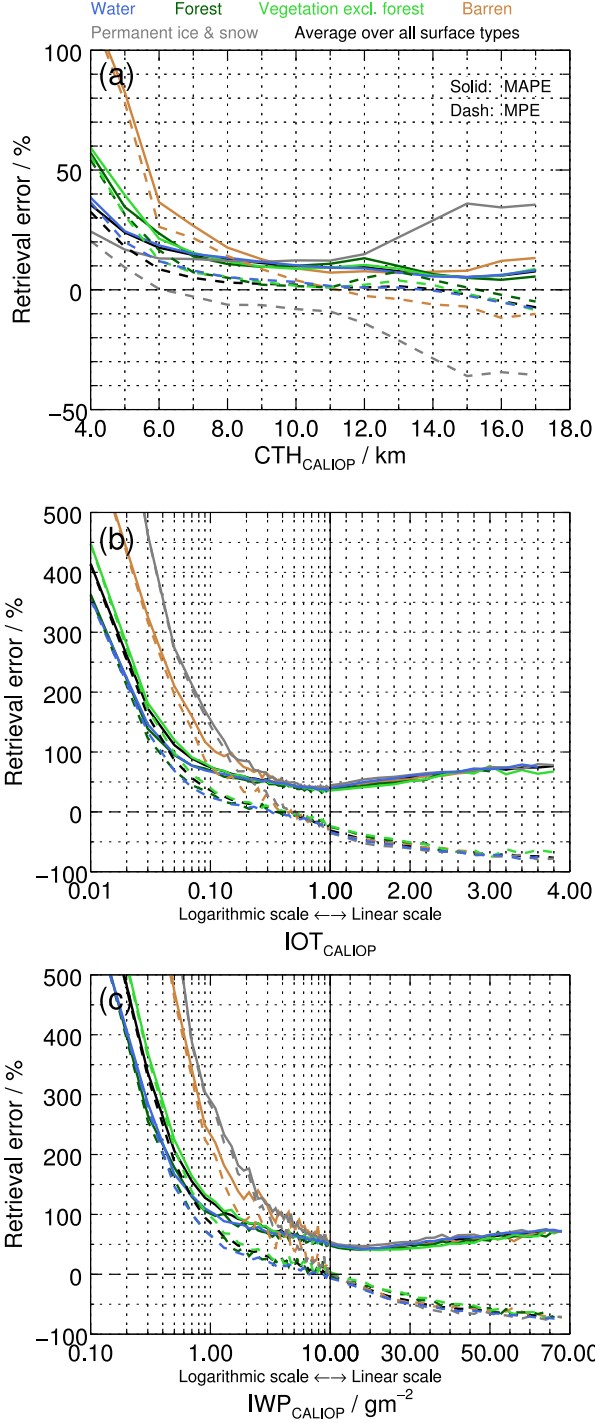

**Figure 4.** The MAPE (solid) and MPE (dash) of the CTH$_{\text{CiPS}}$ (a), IOT$_{\text{CiPS}}$ (b) and IWP$_{\text{CiPS}}$ (c) retrievals as functions of the corresponding reference retrievals from CALIOP. The retrieval errors of CiPS are presented for the five surface type classes introduced in Sect. 4.3.




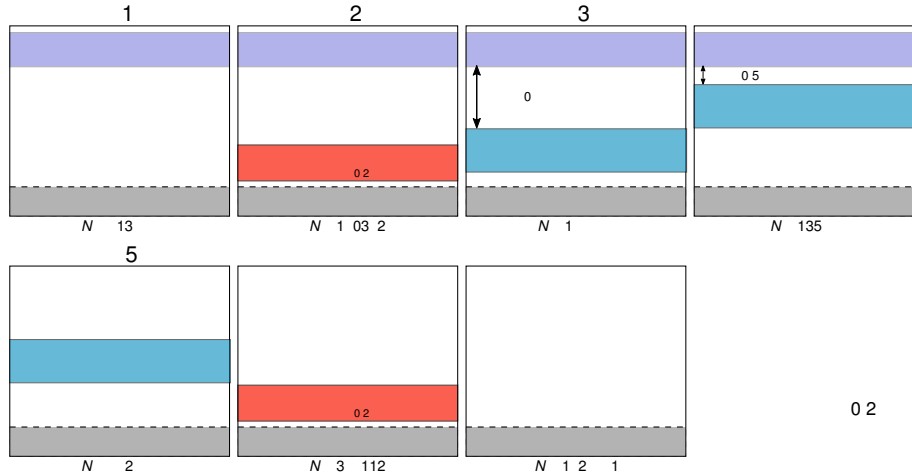

**Figure 5.** Visualisation of the seven vertical cloud-aerosol structure classes. Classes C1-C4 contain transparent cirrus clouds and are used to characterise the CiPS cirrus cloud detection (probability of detection) together with the $CTH_{CiPS}$, $IOT_{CiPS}$ and $IWP_{CiPS}$ retrievals. Classes C5-C7 contain no cirrus clouds and are used to characterise the false alarm rate of the CiPS cirrus cloud detection.

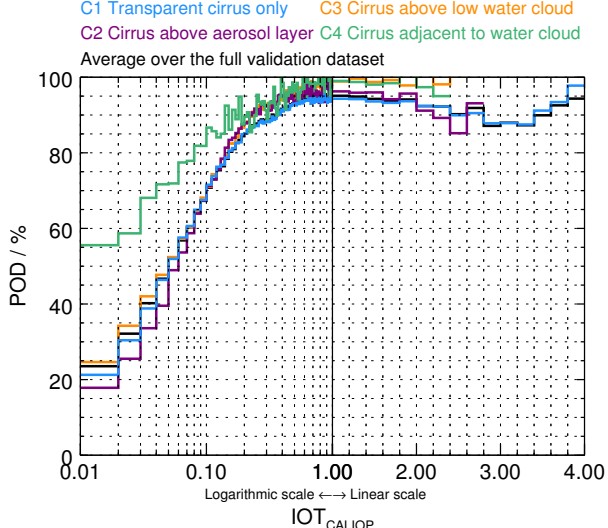

**Figure 6.** The cirrus POD of CiPS as a function of the IOT retrieved by CALIOP for the vertical cloud-aerosol structure classes C1–C4 along with the average POD over the full validation dataset.





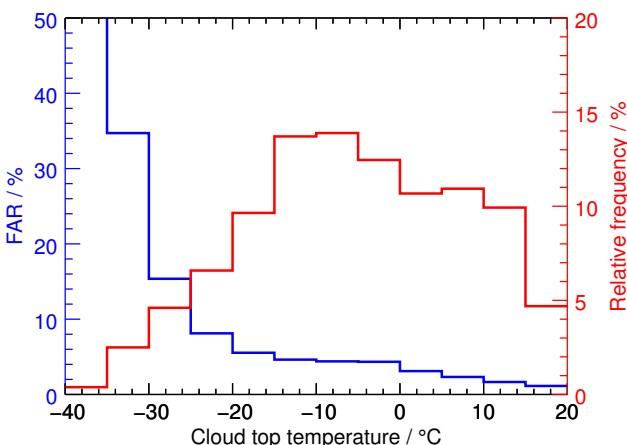

**Figure 7.** The FAR of CiPS for scenes with liquid water clouds (C5) as a function of the corresponding top temperature of the liquid water clouds. Along with the FAR, the relative frequency of occurrence of the different liquid water cloud top temperatures is shown.





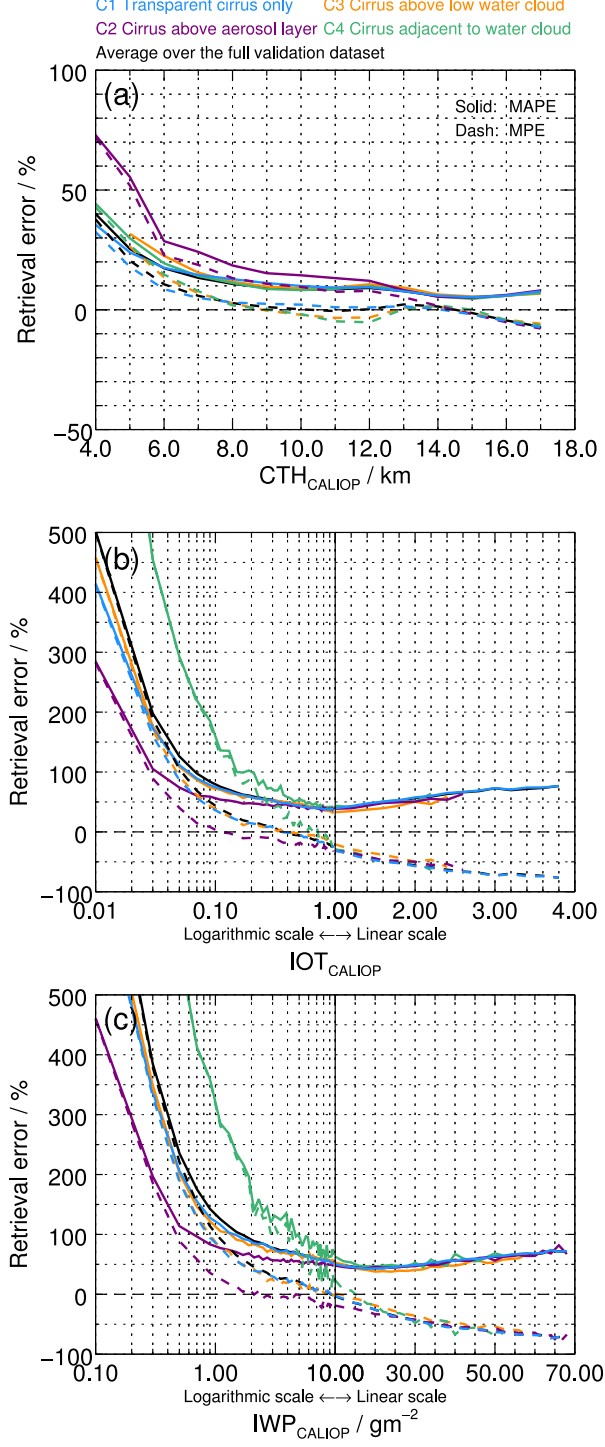

**Figure 8.** The MAPE (solid) and MPE (dash) of the $CTH_{CiPS}$ (a), $IOT_{CiPS}$ (b) and $IWP_{CiPS}$ (c) retrievals as functions of the corresponding reference retrievals from CALIOP. The retrieval errors of CiPS are presented for the four vertical cloud-aerosol structure classes C1-C4 (see Sect. 4.4.1), representing common cloud remote sensing situations, as well as the average retrieval errors for the full validation dataset.







**Figure 9.** (a) 2D histogram showing the MAPE of the $CTH_{CiPS}$ retrievals as a function of $IOT_{CALIOP}$ and the reference CTH retrievals by CALIOP ($CTH_{CALIOP}$). (b) 2D histogram showing the MPE of the $CTH_{CiPS}$ retrievals as a function of $IOT_{CALIOP}$ and $CTH_{CALIOP}$. (c) 2D histogram showing the number of occurrences for different combinations of $IOT_{CALIOP}$ and $CTH_{CALIOP}$. (d) The MAPE of the $CTH_{CiPS}$ retrievals for the different $IOT_{CALIOP}/CTH_{CALIOP}$ combinations as a function of the number of occurrences (each diamond represents two corresponding grid boxes in (a) and (c)).





**Figure 10.** (a) 2D histogram showing the MAPE of the $IOT_{CiPS}$ retrievals as a function of the reference IOT retrievals by CALIOP ($IOT_{CALIOP}$) and $CTH_{CALIOP}$. (b) 2D histogram showing the MPE of the $IOT_{CiPS}$ retrievals as a function of $IOT_{CALIOP}$ and $CTH_{CALIOP}$. (c) 2D histogram showing the number of occurrences for different combinations of $IOT_{CALIOP}$ and $CTH_{CALIOP}$. (d) The MAPE of the $IOT_{CiPS}$ retrievals for the different $IOT_{CALIOP}/CTH_{CALIOP}$ combinations as a function of the number of occurrences (each diamond represents two corresponding grid boxes in (a) and (c)).





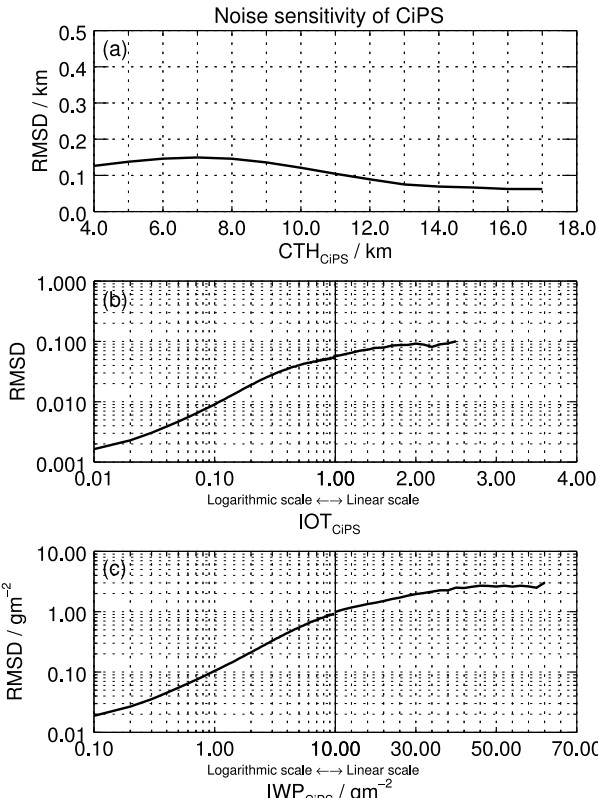

**Figure 11.** The noise sensitivity of the CiPS $CTH_{CiPS}$ (a), $IOT_{CiPS}$ (b) and $IWP_{CiPS}$ (c) retrievals. The noise sensitivity is reported as the RMSD between the CiPS standard retrieval and 100 retrievals where the SEVIRI input data are randomly perturbed within the radiometric noise range of SEVIRI.