# Peer review of "Characterisation of the artificial neural network CiPS for cirrus cloud remote sensing with MSG/SEVIRI"

_Atmospheric Measurement Techniques, 2017_

## Referee Comment (RC1) · Anonymous Referee #1 · 7 Aug 2017

**General comment**

This study describes the performance of the ANN CiPS, used to retrieve cirrus cloud properties from MSG SEVIRI. The assessment includes analyses based on different land cover types, vertical arrangements of clouds and aerosols, retrieved cloud properties and sensitivity to noise in primary observations. It is within the scope of AMT, generally well structured, and the results are adequately presented and explained. Since it can contribute to the overall understanding on the performance, advantages and limitations of ANNs used for cloud properties retrievals, I recommend publication of this study after some minor clarifications and corrections given below.

[Figure]

Specific Comments

Section 3.2

A table would be useful here for better visualizing which input variables were used in each ANN. Since this is provided in Strandgren et al. (2017), the authors could refer to Table 2 of that study.

Section 4

While the OPF flag is an output of CiPS, it is not clear how it was treated in this study; was it just used for excluding opaque cirrus retrievals when characterising other CiPS output? Please clarify.

Section 4.3.2

First paragraph: It would be helpful for the reader if Fig. 5 was also referred here.

Section 4.5

Page 17, lines 8-10: This sentence provides general information on the CiPS training and evaluation data sets. The authors should consider adding this information to Section 2.3, in order to make more clear how the collocation, training and validation data sets are related.

Technical corrections

Page 3, line 27: the word "used" is repeated.

Page 10, line 12: Please replace "...are visualised..." with "...is visualised..."

Page 11, line 3: Please replace "liquid water/aerosol" with "liquid water cloud/aerosol".

Page 13, line 22: Please omit the second "the".

Page 14, line 12: "...liquid water..." should be replaced with "...liquid water clouds..."

Page 21, line 28: Please replace "increases" with "increase".

Page 22, lines 13-14: "... adjacent liquid water clouds..." should be "... adjacent to liquid water clouds..."

---

## Referee Comment (RC2) · Anonymous Referee #2 · 22 Aug 2017

The authors have done a very good job at characterising the performance of CiPS. I am impressed by the depth. The only section where I note some problems, is the final section before the conclusions, on uncertainty estimates and propagation. That one may be harder to get right than it seems and is not essential for the paper, so perhaps the best option here is to skip it. Regardless of that section, a list of suggestions for a minor revision follows below.

page 1, line 3: replace "implemented" by "modelled". Only nature can implement physics!

page 1, line 20: what do you mean by "physical implementation"? The implementation

of the physics in the model?

page 2, lines 11-12: the formulation here is a bit sloppy. The radiometer itself does not have a spatial resolution or a vertical component. The spatial resolution comes from either a scanning mechanism or a detector array. The ability to resolve vertical information from passive sensors depends on the availability of multiple channels with different weighting functions, but not in clouds. I would rephrase this sentence as: "Imaging radiometers typically view a large area (by scanning or otherwise) to observe complete cloud systems, but a passive infrared sensor cannot resolve cloud features vertically and has a limited sensitivity to thin and sub-visual (visible optical thickness < 0.03) cirrus clouds." or similar.

page 2, line 16: the active nature does not intrinsically lead to a poor spatial coverage, but is a consequence of other design considerations (scanning radars exist in space and certainly on the ground). I would replace "leads to ... between orbits" by "those sensors have a small footprint and observe only at nadir, which leads to a poor spatial coverage."

page 2, lines 19-31: ANNs can indeed exploit collocations to train a retrieval, but that property is by no means unique to ANNs. Any type of machine learning can do, whether it is a basic linear regression, a neural network, a support vector machine, or others. You need to add a couple of lines here, pointing out that a collocation database can be constructed with most pairs of orbits and that this can be used to train a retrieval. For cloud retrievals, ANNs have proven to work quite well to perform this training in practice.

page 3, line 22: replace "brightness temperature" by "brightness temperatures" — you're using multiple channels

page 4, lines 6–7: you could add a small figure showing a map of the SEVIRI disc.

page 4, line 12: those radiometric noise levels are design specifications and not actual measured noise levels. Actual noise levels can be derived from on-board level-1 measurements by looking at the dark corners of the disc and then propagating the uncertainty in counts through to radiance, reflectance, and brightness temperature units. If you are going to use those noise levels for uncertainty estimates you need to be aware they can be quite far from actual noise. If you are not using them I would not report them, as it just propagates a number that many people are using incorrectly.

page 5, line 3: IWP is not a layer product but a column-integrated product. Should this perhaps be IWC or pIWP (partial column IWP)?

page 9, Figure 1: although I think the figure looks beautifully crafted I'm not sure if it's the most effective way to visualise the weights. At least, the colour for OPF is too similar to the colour for IOT&IWP. But in general, I believe the information would be easier to read in a tabular format, with 4 columns (for the products) and 17 rows (for the input variables), writing the weight as a number in each table cell, and perhaps colour-coding by the value. The downside of the figure is that the connecting line appears to give an implied meaning to an essentially meaningless ordering; it may be hard to read values when they are very close to each other; and the relatively long tick labels necessitate alternating them between the bottom and the top, which adds to the confusion (if the authors rotate the labels they could have them all at the bottom). A table might work better. Yet another way might be a flow diagram, where the seventeen inputs would be written below each other and the thickness of the line connecting to each input would be proportional to the weight (those can again be labelled), although I'm not sure how cluttered the visualisation might become when four output "flows" are visualised in the same diagram.

page 9, Figure 1: why are all weights positive? Is this a standard property of the ANN, did the authors impose it, or is it a coincidence? In particular for the DOY_SIN and DOY_COS, which after all can take either positive or negative value, it is not obvious to me why it should be.

page 10, Figure 2 / page 11, lines 9-10: is there any permanent ice & snow in the SEVIRI disc? The line for "permanent ice and snow" looks rather noisy. Is that just the strip of Greenland barely in the field of view? How many pixels are those? The POD appears remarkably good over permanent ice and snow but I wonder how significant the results are.

page 10, line 7: Are all barren surfaces in the field of view bright, hot deserts? I wonder if there are any dark barren surface types to test against? Maybe Iceland? Might that show a better performance?

page 11, lines 14-16: I understand why mixed phase or supercooled liquid clouds complicate the cirrus detection, but why is this (independently) the case for temperature inversions, that should be a boundary layer phenomenon anyway?

page 15, line 19: I'm confused by the reference to Strandgren et al. 2017 here. Surely those are the classes that have been discussed in the previous paragraph of the present paper? Why the reference?

page 15, line 28: 97%... is this shown? If yes, where? If not, please state that it is not shown so the reader does not needlessly look for a figure showing this.

page 17, line 9-10: this is not accurate. CALIOP flies in a sun-synchronous orbit and thus only samples a very limited part of the diurnal cycle. The statement that it represents the "natural" distribution is too strong.

page 17, line 15: replace "error" by "uncertainty". You cannot retrieve an error; the error is the difference between truth and retrieval. The uncertainty is a statistical estimate of the distribution of errors. The uncertainty can be estimated, the error is only known in an artificial scenario.

page 19, line 25-27: again, the authors should be aware that those are most likely not actually measured noise levels, but rather design specifications. If the authors are sure that the former is the case this should be clearly stated, because the design design

specification for radiometers is often misinterpreted as an actual noise level.

page 19, line 32: personal communication with a large institute seems somewhat unusual, is there a name attached to this person?

page 20, line 1-2: I think the authors need to add a caveat that although they obtain a number that varies per-pixel, this is in fact not a metrologically traceable per-pixel uncertainty. Determining the latter is possible (EUMETSAT and others are part of an effort to do this for MVIRI, see www.fiduceo.eu) but a lot of work. As currently phrased, this paragraph is at risk of misleading readers into thinking there are true per-pixel uncertainty estimates.

page 20, line 17-18: the ANN is essentially a set of equations, all of which are differentiable. Therefore, it should be possible to directly apply the Law of Propagation of Uncertainties. Have the authors considered calculating this propagation directly, instead of via an ensemble?

page 21, line 10: what is the resolution / digitisation level of SEVIRI at those brightness temperatures?

page 22, line 17: although I believe this to be accurate, I am not convinced the evidence presented by the authors is sufficient to show this. I suspect it may actually be smaller.

page 30, figure 5: I opened this figure with two different software packages but in both cases it seems something is wrong. I see only panels 1, 2, 3, and 5 labelled, at the bottom I see "N 13", "N 1 03 2", "N 1", "N 135", "N 2", with varying amounts of whitespace. It appears some labels have gone missing.

page 30, figure 6: I find it slightly confusing to see "transparent cirrus" with IOT up to (and possibly exceeding) 4. Perhaps a different word would be more suitable (but I don't feel too strongly about this).

page 33, figure 9: There is some evidence that 2D-histograms with hexagonal binning are superior to those with rectangular binning; the authors may wish to search

the internet for the keyword "hexplot", try to show the data or a hexagonal grid, and judge for themselves. An article illustrating why it may be superior can be found at http://www.meccanismocomplesso.org/en/hexagonal-binning/

---

## Referee Comment (RC3) · Anonymous Referee #3 · 31 Aug 2017

General comments

This paper exploits characterization of CiPS (cirrus cloud detection and property retrieval) that is based on the artificial neural network (ANN). ANN is known to perform well in statistical sense if it is well trained. ANN's performance relies on the architecture (e.g., selection of input variables) and training data. Since there is no physics in the ANN modeling, a thorough characterization is important. I found this paper is interesting. Presented results are useful for interpreting the cirrus remote sensing using similar techniques. In my view, this paper is well suitable to be published in AMT. The manuscript is generally well written. I recommend that this paper is published with

minor revisions. There are several suggestions for revisions as described below.

1. (Section 3.2 or somewhere) IR measurements are sensitive to atmospheric temperature and humidity profiles, as well. CiPS uses surface temperature and latitude and DOY as input. Although they could provide some information about atmospheric profile, I believe it is better to include temperature and humidity profiles. Variability of humidity is particularly large, and I think that surface temperature and latitude and DOY are not enough to model that variability. If atmospheric profile is included in the input, cirrus detection and retrieval can be improved, and water vapor channels can become more important.

2. For CTH, the percentage error is not very comprehensive. Error in CTH scale (unit in km) is more comprehensive.

3. ANN may output multiple variables. Why two ANNs are constructed to estimate CTH and IOT separately? I guess one reason is that by doing so, sensitivities to input can be investigated as shown in Fig. 1. Is there any reason concerning to retrieval accuracy?

Specific comments

Page 1, line 10, "thin": How thin is it?

Page 8, line 17: Results presented in Section 4.2 are interesting and useful. I am just wondering how the weights are normalized. Is variability of every input variable normalized?

---

## Author Comment (AC1) · 15 Sep 2017

We thank the reviewer for taking the time to read and review our manuscript. Each comment from the reviewer (roman style) is listed below along with the corresponding reply from the authors (in italic font style) as well as possible changes in the manuscript (in blue italic font style).

**Specific comments**
Section 3.2
A table would be useful here for better visualizing which input variables were used in

each ANN. Since this is provided in Strandgren et al. (2017), the authors could refer to Table 2 of that study.

*We thank the reviewer for the good suggestion, a reference to Strandgren et al. (2017) has been added in the end of Sect. 3.2: "please see Table 2 in Strandgren et al. (2017) for a tabular overview of all input variables."*

Section 4
While the OPF flag is an output of CiPS, it is not clear how it was treated in this study; was it just used for excluding opaque cirrus retrievals when characterising other CiPS output? Please clarify.

*For the input variable importance analysis, the CiPS opacity flag (OPF$_{CiPS}$) is treated just like the other CiPS output variables. For the surface type as well as for the vertical cloud-aerosol structure analysis the OPF-flag is not used at all. OPF$_{CALIOP}$, rather than OPF$_{CiPS}$, is used to identify and exclude opaque retrievals since CALIOP is more accurate. For opaque retrievals, neither the surface type nor the vertical cloud-aerosol structure analysis makes sense since we don't know about possible cloud/aerosol layers below the cirrus. Also when the retrieval errors are investigated as a function of IOT$_{CALIOP}$ and CTH$_{CALIOP}$ only transparent cirrus retrievals, as defined by OPF$_{CALIOP}$, are included. Only for the noise sensitivity analysis, where no CALIOP data are used, OPF$_{CiPS}$ is used to exclude opaque IOT$_{CiPS}$ and IWP$_{CiPS}$ retrievals.*

*For the input variable importance and the noise sensitivity analysis we consider it clear how OPF$_{CiPS}$ was used. But for the other parts of the characterisation, we agree with the reviewer that there is room for clarification. The following changes in the manuscript have been made to clarify this aspect: In Sect. 4.3.2 we now write "The goal of the opacity flag retrieved by CiPS is to detect cirrus clouds that are opaque, i.e. where the vertical structure below the cirrus is unknown for CiPS/CALIOP.*

*Consequently, the opacity flag of CiPS is not characterised for the different surface types as it cannot be ruled out that there are no liquid water clouds or aerosol layers with AOT $> 0.2$ below an opaque cirrus. Please note that the more accurate opacity flag of CALIOP is used to identify profiles with opaque cirrus clouds that are excluded from the analysis, as explained in Sect. 4.4.1." In Sect. 4.4 we now write: "Again, only the CiPS quantities $CCF_{CiPS}$, $CTH_{CiPS}$, $IOT_{CiPS}$ and $IWP_{CiPS}$ are characterised for the different vertical cloud-aerosol structures. The $OPF_{CiPS}$ is excluded from the analysis since its goal is to detect cirrus clouds where the vertical structure below the cirrus cannot be resolved by CALIOP. Opaque cirrus clouds are identified and excluded using the opacity flag of CALIOP as described in the following section.". In Sect. 4.5 the sentence "To remove any effects from different vertical cloud-aerosol structures, again only those profiles with transparent cirrus clouds and possible faint aerosols (AOT $\leq 0.2$) are used (class C1 in Sect. 4.4.1)" has been rephrased as: "To remove any effects from different vertical cloud-aerosol structures, again only those profiles with transparent cirrus clouds and possible faint aerosols (AOT $\leq 0.2$)* **as defined by CALIOP L2 data** *are used (class C1 in Sect. 4.4.1)".*

Section 4.3.2
First paragraph: It would be helpful for the reader if Fig. 5 was also referred here.

*A reference to Fig. 5 has been added.*

Section 4.5
Page 17, lines 8-10: This sentence provides general information on the CiPS training and evaluation data sets. The authors should consider adding this information to Section 2.3, in order to make more clear how the collocation, training and validation data sets are related.

*This information has been added as proposed by the reviewer. The last part of Sect.*

*2.3 now reads as follows: "This dataset was originally used to validate CiPS and contains 5 million collocations collected over a time period of almost 6 years (April 2007 to January 2013). This represents a random subset containing 10 % of all quality-screened collocations of CiPS input data and CALIOP cirrus cloud properties obtained during this time period. The remaining 90 % of the collocations were used to develop and train CiPS. Hence, the collocation dataset, as well as the training datasets used to develop CiPS, do to some extent (limited by the sun-synchronous orbit of CALIPSO) represent the natural distribution of cirrus clouds and cirrus cloud properties. A detailed description of the collocation dataset can be found in Strandgren et al. (2017), where it is referred to as the internal validation dataset."*

**Technical corrections**

Page 3, line 27: the word "used" is repeated.

*Revised*

Page 10, line 12: Please replace "... are visualised..." with "... is visualised..."

*Revised*

Page 11, line 3: Please replace "liquid water/aerosol" with "liquid water cloud/aerosol".

*Revised*

Page 13, line 22: Please omit the second "the".

*Revised*

Page 14, line 12: "...liquid water..." should be replaced with "...liquid water clouds..."

*Revised*

Page 21, line 28: Please replace "increases" with "increase".

*Revised*

Page 22, lines 13-14: "...adjacent liquid water clouds..." should be "...adjacent to liquid water clouds..."

*Revised*

---

## Author Comment (AC2) · 15 Sep 2017

We thank the reviewer for taking the time to read and review our manuscript. Each comment from the reviewer (roman style) is listed below along with the corresponding reply from the authors (in italic font style) as well as possible changes in the manuscript (in blue italic font style).

**General comments**

1. (Section 3.2 or somewhere) IR measurements are sensitive to atmospheric temperature and humidity profiles, as well. CiPS uses surface temperature and latitude

and DOY as input. Although they could provide some information about atmospheric profile, I believe it is better to include temperature and humidity profiles. Variability of humidity is particularly large, and I think that surface temperature and latitude and DOY are not enough to model that variability. If atmospheric profile is included in the input, cirrus detection and retrieval can be improved, and water vapor channels can become more important.

*We agree with the reviewer that including vertical humidity and temperature profiles as input data would improve the performance of CiPS. This is also something that was addressed in our previous manuscript that was recently accepted for publication in AMT, where the development and validation of CiPS is presented (AMTD version available here: https://www.atmos-meas-tech-discuss.net/amt-2017-64/). We chose not to include any vertical profiles in order to keep the computational costs down. By including vertical profiles of temperature and humidity we would increase the number of input neurons and consequently the time required to train and use CiPS considerably. We have added some lines about this in the concluding section of this manuscript: "In general auxiliary data like surface type flags and day of the year are shown to have a comparably small relative importance and for future developments within this field, surface emissivities as well as vertical humidity and temperature profiles would probably prove more useful. Using vertical profiles as input data would increase the computational costs though."*

2. For CTH, the percentage error is not very comprehensive. Error in CTH scale (unit in km) is more comprehensive.

*The reviewer is right, probably it would be more reasonable to use a relative error measure (%) for IOT and IWP, but an absolute error (km) for the CTH. However, we would prefer to stick to the relative error for the CTH for consistency purposes, since this is also used in the first manuscript about CiPS (see above). We think that it would*

*be less confusing if we stick to the same error measure for the same variable in the two manuscripts.*

3. ANN may output multiple variables. Why two ANNs are constructed to estimate CTH and IOT separately? I guess one reason is that by doing so, sensitivities to input can be investigated as shown in Fig. 1. Is there any reason concerning to retrieval accuracy?

*In principle the retrieval accuracy should increase if the output variables are retrieved with separate ANNs. But our reason for doing so is related to the fact that the CTH can be retrieved by CALIOP (our training reference data source) for all cirrus/ice clouds independent of the thickness, whereas the IOT/IWP can only be retrieved for cirrus clouds with an optical thickness up to about 3 (usually less) due to the saturation of the laser beam. Consequently we can only train the IOT/IWP ANN with cirrus cloud retrievals where CALIOP could fully penetrate the cirrus cloud (transparent cirrus). The CTH ANN however, can be trained with all cirrus/ice cloud retrievals independent of transparency/opacity. To summarise, the reason to use two ANNs is that we use different training datasets. More details about this is available in the first manuscript about CiPS, referred to above.*

**Specific comments**
Page 1, line 10, "thin": How thin is it?

*It refers to an optical thickness less than approx. 0.3. This information has been added in the revised manuscript.*

Page 8, line 17: Results presented in Section 4.2 are interesting and useful. I am just wondering how the weights are normalized. Is variability of every input variable

normalized?

*This part has been extended and clarified and it is now clearly written how the relative importance of the single input variables is obtained. This part now reads as follows: "The importance of an input variable can be estimated as the euclidean length of the vector holding all weights that connect that input neuron with the hidden neurons in the first hidden layer (LeCun et al., 1990). The importance (or total weight) of an input variable $i$ is thus calculated as $W_i = \sqrt{w_{i,1}^2 + w_{i,2}^2 + \ldots + w_{i,N}^2}$, where $w_{i,1}$ to $w_{i,N}$ are the single weights connecting input variable $i$ with the $N$ neurons in the first hidden layer. Figure 1 shows the relative importance of the 18 input variables used by CiPS. The relative importance of all input variables is calculates as $W_i^* = 100\,\% \cdot W_i / (W_1 + W_2 + \ldots + W_{18})$ for the respective ANNs such that the sum of the relative importance across all input variables adds up to 100\,% for each ANN."*

---

## Author Comment (AC3) · 18 Sep 2017

We thank the reviewer for reading and reviewing our manuscript and appreciate the kind and constructive feedback that helped us to improve the quality of the manuscript. Each comment from the reviewer (roman style) is listed below along with the corresponding reply from the authors (in italic font style) as well as possible changes in the manuscript (in blue italic font style).

**Specific comments**
page 1, line 3: replace "implemented" by "modelled". Only nature can implement

physics!

*Revised*

page 1, line 20: what do you mean by "physical implementation"? The implementation of the physics in the model?

*Yes, that is what we mean. The sentence has been rephrased accordingly: "... even though the retrieval methods differ in the implementation of physics in the model, ..."*

page 2, lines 11-12: the formulation here is a bit sloppy. The radiometer itself does not have a spatial resolution or a vertical component. The spatial resolution comes from either a scanning mechanism or a detector array. The ability to resolve vertical information from passive sensors depends on the availability of multiple channels with different weighting functions, but not in clouds. I would rephrase this sentence as: "Imaging radiometers typically view a large area (by scanning or otherwise) to observe complete cloud systems, but a passive infrared sensor cannot resolve vertical cloud features and has a limited sensitivity to thin and sub-visual (visible optical thickness $<$ 0.03) cirrus clouds." or similar.

*We thank the reviewer for pointing this out. The sentence has been rephrased as kindly suggested by the reviewer: "Imaging radiometers typically view a large area (by scanning or otherwise) to observe complete cloud systems, but a passive infrared sensor cannot resolve cloud features vertically and has a limited sensitivity to thin and sub-visual (visible optical thickness $<$ 0.03) cirrus clouds".*

page 2, line 16: the active nature does not intrinsically lead to a poor spatial coverage, but is a consequence of other design considerations (scanning radars exist in space

and certainly on the ground). I would replace "leads to ... between orbits" by "those sensors have a small footprint and observe only at nadir, which leads to a poor spatial coverage."

*Again we thank the reviewer for pointing this out. Again we have rephrased the sentence according to the reviewers suggestion: "However, those sensors have a small footprint and observe only at nadir, which leads to a poor spatial coverage.".*

page 2, lines 19-31: ANNs can indeed exploit collocations to train a retrieval, but that property is by no means unique to ANNs. Any type of machine learning can do, whether it is a basic linear regression, a neural network, a support vector machine, or others. You need to add a couple of lines here, pointing out that a collocation database can be constructed with most pairs of orbits and that this can be used to train a retrieval. For cloud retrievals, ANNs have proven to work quite well to perform this training in practice.

*We agree with the reviewer and this has been rewritten with more general terms regarding the exploitation of collocations. This part now reads as follows: "Combining the advantages of satellite sensors operating in different orbits is more challenging, as they observe given scenes at different times from possibly different perspectives. Nevertheless, the information from available sensor collocations can be used to learn relationships between different sets of observations, e.g. through machine learning. For cloud remote sensing, artificial neural networks (ANNs) have proven to be a powerful tool for this".*

page 3, line 22: replace "brightness temperature" by "brightness temperatures" — you're using multiple channels

*Revised*

page 4, lines 6–7: you could add a small figure showing a map of the SEVIRI disc.

*A reference to Fig. 2 has been added here: "The spatial coverage of SEVIRI can be seen in Fig. 2"*

page 4, line 12: those radiometric noise levels are design specifications and not actual measured noise levels. Actual noise levels can be derived from on-board level-1 measurements by looking at the dark corners of the disc and then propagating the uncertainty in counts through to radiance, reflectance, and brightness temperature units. If you are going to use those noise levels for uncertainty estimates you need to be aware they can be quite far from actual noise. If you are not using them I would not report them, as it just propagates a number that many people are using incorrectly.

*After consulting EUMETSAT both before the manuscript was submitted and again because of this comment, we have to say that we disagree with the reviewer. Table 4 in EUMETSAT (2007) shows the measured radiometric accuracy of the cold MSG-2/SEVIRI channels. The rightmost column shows the design specifications at given reference temperatures. The second and third columns show the noise estimates derived from measurements of the internal black body calibration target at black body temperatures around 280-300 K (ambient calibrations) and 300-320 K (heated calibrations), but re-scaled to the given reference temperatures associated with the design specifications. In this study we have used the "ambient calibrations", which are derived from actual measurements. In the revised manuscript, we have however clarified that those are estimates/indicators and not necessarily representative for all SEVIRI retrievals: "Estimates of the radiometric noise levels of the SEVIRI thermal channels can be derived from measurements of the internal black body calibration target and are reported as... . Please note that the reported noise levels are estimators/indicators and not necessarily representative for any given SEVIRI observation. For a statistical*

*analysis however, those estimates are sufficient."*

*EUMETSAT: Typical Radiometric Accuracy and Noise for MSG-1/2, https://www.eumetsat.int/website/ home/Data/Products/Calibration/MSGCalibration/index.html, 2007.*

page 5, line 3: IWP is not a layer product but a column-integrated product. Should this perhaps be IWC or pIWP (partial column IWP)?

*As the reviewer suggests, partial column IWP is a better definition since the layer IWP reported in the CALIOP L2 cloud layer product reports the integrated ice water content between the base heigh and top height of a given cloud layer classified as ice. "partial column" has been added to the manuscript.*

page 9, Figure 1: although I think the figure looks beautifully crafted I'm not sure if it's the most effective way to visualise the weights. At least, the colour for OPF is too similar to the colour for IOT&IWP. But in general, I believe the information would be easier to read in a tabular format, with 4 columns (for the products) and 17 rows (for the input variables), writing the weight as a number in each table cell, and perhaps colour-coding by the value. The downside of the figure is that the connecting line appears to give an implied meaning to an essentially meaningless ordering; it may be hard to read values when they are very close to each other; and the relatively long tick labels necessitate alternating them between the bottom and the top, which adds to the confusion (if the authors rotate the labels they could have them all at the bottom). A table might work better. Yet another way might be a flow diagram, where the seventeen inputs would be written below each other and the thickness of the line connecting to each input would be proportional to the weight (those can again be labelled), although I'm not sure how cluttered the visualisation might become when four output "flows" are visualised in the same diagram.

*This figure was indeed not straight-forward to craft with many input variables and long labels. The fact that the ordering, from left to right, might be expected to have some meaning is also true, but something that we did not think about. We have adapted the figure according to the reviewers constructive feedback and the relative importance is now visualised in a colour-coded tabular format. We think that the new figure presents the results in a much clearer way.*

page 9, Figure 1: why are all weights positive? Is this a standard property of the ANN, did the authors impose it, or is it a coincidence? In particular for the DOY_SIN and DOY_COS, which after all can take either positive or negative value, it is not obvious to me why it should be.

*The single weights are both positive and negative, but Fig. 1 is based on the euclidean length of the vector of weights connected to the corresponding input neurons for each ANN (page 8, line 17). This prevents negative values since the total weight $W_i$ of an input variable $i$ is given by $W_i = \sqrt{w_{i,1}^2 + w_{i,2}^2 + \ldots w_N^2}$, where $w_{i,1}$ to $w_{i,N}$ are the single weights (positive and negative) connecting input variable $i$ with the $N$ neurons in the first hidden layer. The derivation of the importance/total weight of the input variables as well as the conversion to relative importance has been clarified in the revised manuscript:"The importance (or total weight) of an input variable $i$ is thus calculated as $W_i = \sqrt{w_{i,1}^2 + w_{i,2}^2 + \ldots + w_{i,N}^2}$, where $w_{i,1}$ to $w_{i,N}$ are the single weights connecting input variable $i$ with the $N$ neurons in the first hidden layer. Figure 1 shows the relative importance of the 18 input variables used by CiPS. The relative importance of all input variables is calculates as $W_i^* = 100\% \cdot W_i/(W_1 + W_2 + \ldots + W_{18})$ for the respective ANNs such that the sum of the relative importance across all input variables adds up to 100% for each ANN.". We realise that the use of the term "weight" for both the single weights connecting the neurons and the overall importance of an input*

*variable itself might be confusing, this has been revised and the term "importance" is now used for the total weight of an input variable throughout the manuscript.*

page 10, Figure 2 / page 11, lines 9-10: is there any permanent ice & snow in the SEVIRI disc? The line for "permanent ice and snow" looks rather noisy. Is that just the strip of Greenland barely in the field of view? How many pixels are those? The POD appears remarkably good over permanent ice and snow but I wonder how significant the results are.

*As the reviewer implies, the amount of permanent ice & snow is limited in the SEVIRI disc. The two main sources are Greenland and Antarctica. There are also a few pixels of permanent ice & snow at some high altitudes in mountain ranges like the Alps, Andes and Scandes. In total, permanent ice & snow constitutes just 0.3 % of the SEVIRI disc, but with nearly six years of collocations we still have 47 000 CALIOP-SEVIRI collocations with transparent cirrus (without liquid water clouds and AOT $\leq 0.2$) over permanent ice & snow. Hence we consider the results to be significant. Please note that we have the same number of collocations with solely transparent cirrus also over forest and 36 000 collocations over barren. To clarify how many points are used calculate the POD over the different surface types, the number of collocations for the largest and smallest groups has been added to Sect. 4.3.2: "In total approx. 600 000 such collocations are available in the collocation dataset, with the largest number of occurrences over water (360 000) and the smallest number over barren (36 000)."*

page 10, line 7: Are all barren surfaces in the field of view bright, hot deserts? I wonder if there are any dark barren surface types to test against? Maybe Iceland? Might that show a better performance?

*Yes, nearly all barren surfaces are bright deserts (approx. 99.3 %). Some areas of barren are found in the Andes and Iceland, but with such an unbalanced distribution,*

*we are sceptical that a comparison between bright/warm and dark/cold barren is meaningful. Instead the following sentences have been added to Sect. 4.3.1 in order to clarify that the results are rather representative for retrievals over desert than over barren in general: "Please note that the class barren is composed mostly of bright desert surfaces in the SEVIRI disc. Hence the results presented for barren in this section are mostly representative for retrievals over desert and only to a very limited extend for retrievals over other types of barren present in e.g. the Andes and Iceland."*

page 11, lines 14-16: I understand why mixed phase or supercooled liquid clouds complicate the cirrus detection, but why is this (independently) the case for temperature inversions, that should be a boundary layer phenomenon anyway?

*In the polar regions temperature inversions are frequent and can make the cloud top appear warmer than the snow/ice covered surface and reduce the detection of low clouds. However, this effect is relevant for ice clouds only in the cold polar atmospheres (in mid-latitudes liquid water cloud detection is affected by this problem). To clarify, we rephrased: "Furthermore, mixed phase clouds or supercooled liquid water layers above ice layers in the polar regions (Mioche et al., 2015; Verlinde et al., 2007; Shupe et al., 2006) may also reduce the POD as CiPS requires the water to be frozen to be classified as a cirrus. Moreover, temperature inversions, frequent in these areas (Wetzel and Brümmer, 2011), can make the cloud top of low ice clouds (Devasthale et al. 2011) appear warmer than the snow/ice covered surface and thus reduce their detection (Wilson et al., 1993; Gao et al., 1998)."*

*Wetzel, C. and Brümmer, B.: An Arctic inversion climatology based on the European Centre Reanalysis ERA-40, Meteor. Z., 20, 589–600, 2011.*

*Devasthale, A., Tjernström, M., Karlsson, K.-G., Thomas, M. A., Jones, C., Sedlar, J., and Omar, A. H.: The vertical distribution of thin features over the Arctic analysed from*

*CALIPSO observations, Tellus B, 63, 77–85, 2011.*

*Wilson, L. D., Curry, J. A., and Ackerman, T. P.: Satellite Retrieval of Lower-Tropospheric Ice Crystal Clouds in the Polar Regions, J. Climate, 6, 1467–1472, 1993.*

*Gao, B.-C., Han, W., Tsay, S. C., and Larsen, N. F.: Cloud Detection over the Arctic Region Using Airborne Imaging Spectrometer Data during the Daytime, J. Appl. Meteor., 37, 1421–1429, 1998.*

page 15, line 19: I'm confused by the reference to Strandgren et al. 2017 here. Surely those are the classes that have been discussed in the previous paragraph of the present paper? Why the reference?

*What we wanted to say is that the probability of detection (POD) for all cirrus clouds and not only those that fit one of the classes C1-C4 is the POD initially presented in Strandgren et al. (2017), where liquid water clouds and aerosol layers below the cirrus were not considered. The black line in Fig. 6 is consequently identical to the POD-curve for CiPS in Strandgren et al. (2017). Since the reference is not essential we have removed it to avoid confusion.*

page 15, line 28: 97%... is this shown? If yes, where? If not, please state that it is not shown so the reader does not needlessly look for a figure showing this.

*No, this is not shown. It is simply related to the false alarm rate (FAR) which we throughout the paper present as single numbers i.e. without plots ($100\,\% - \mathrm{FAR} = 100\,\% - 3.2\,\% \approx 97\,\%$). This sentence has been clarified to avoid confusion and now reads as follows: "For scenes with clear sky (C7) or thicker aerosol layers (C6) CiPS has a FAR of 3.2 %, meaning that it correctly classifies close to 97 % of such scenes as cirrus free (not further shown here)."*

page 17, line 9-10: this is not accurate. CALIOP flies in a sun-synchronous orbit and thus only samples a very limited part of the diurnal cycle. The statement that it represents the "natural" distribution is too strong.

*We agree that this was sloppy formulated. This part now reads as follows: "As mentioned in Sect. 2.3, both the collocation dataset as well as the training datasets used to train CiPS consist of a random subset of CALIOP data collected over a time period of almost 6 years and do to some extent (limited by the sun-synchronous orbit of CALIPSO) represent the natural distribution of IOT and CTH frequencies and their combinations."*

page 17, line 15: replace "error" by "uncertainty". You cannot retrieve an error; the error is the difference between truth and retrieval. The uncertainty is a statistical estimate of the distribution of errors. The uncertainty can be estimated, the error is only known in an artificial scenario.

*Revised. To further clarify this sentence, we rephrase: "In the following we investigate the retrieval errors (MAPE and MPE) of CiPS with respect to CALIOP for different combinations of $IOT_{CALIOP}$ and $CTH_{CALIOP}$ in order to quantify and characterise the CiPS retrieval uncertainties that can be expected in general for different cirrus cloud types."*

page 19, line 25-27: again, the authors should be aware that those are most likely not actually measured noise levels, but rather design specifications. If the authors are sure that the former is the case this should be clearly stated, because the design design specification for radiometers is often misinterpreted as an actual noise level.

*Revised. Please see our response to the comment above regarding page 4, line 12.*

page 19, line 32: personal communication with a large institute seems somewhat unusual, is there a name attached to this person?

*Yes, this information has been added in the revised manuscript.*

page 20, line 1-2: I think the authors need to add a caveat that although they obtain a number that varies per-pixel, this is in fact not a metrologically traceable per-pixel uncertainty. Determining the latter is possible (EUMETSAT and others are part of an effort to do this for MVIRI, see www.fiduceo.eu) but a lot of work. As currently phrased, this paragraph is at risk of misleading readers into thinking there are true per-pixel uncertainty estimates.

*This is a good point. The following sentence has been added to the revised manuscript: "Please note that those are no metrologically traceable per-pixel noise estimates, instead all noise estimates are directly related via the observed brightness temperatures to the overall noise estimates of the single channels reported in the second column in Table 1"*

page 20, line 17-18: the ANN is essentially a set of equations, all of which are differentiable. Therefore, it should be possible to directly apply the Law of Propagation of Uncertainties. Have the authors considered calculating this propagation directly, instead of via an ensemble?

*The reviewer this right that this would be a more straight forward approach that would have saved computational time since only one uncertainty propagation instead of 100 perturbed retrievals would be required for each sample. However this is something*

*that we did not think about. We thank the reviewer for pointing this out and will keep it in mind for future applications, but since both approaches should yield similar results (given that our ensemble of 100 perturbations is large enough) we stick to the current approach in this manuscript.*

page 21, line 10: what is the resolution / digitisation level of SEVIRI at those brightness temperatures?

*For an example observation by SEVIRI we calculate the digitisation level to be around 0.10–0.20 K for the six SEVIRI channels used as input for CiPS at the corresponding brightness temperatures typically observed for cirrus clouds (see third column of Table 1).*

page 22, line 17: although I believe this to be accurate, I am not convinced the evidence presented by the authors is sufficient to show this. I suspect it may actually be smaller.

*We assume that the comment is related to the fact that the reviewer questioned whether the reported noise levels were the instrument specifications or actual measured noise. We are confident that the noise levels are derived from measurements and we therefore claim this to be a valid statement. However we have rephrased the sentence in order to clarify that it is an estimation: "Only a small fraction of the retrieval errors ($\approx 10\,\%$) is estimated to stem from radiometric noise in the SEVIRI data."*

page 30, figure 5: I opened this figure with two different software packages but in both cases it seems something is wrong. I see only panels 1, 2, 3, and 5 labelled, at the bottom I see "N 13", "N 1 03 2", "N 1", "N 135", "N 2", with varying amounts of whitespace. It appears some labels have gone missing.

*The reviewer is right, somehow this figure got broken when the original PDF was converted to the AMTD manuscript and alot of text and labels are missing. We are sorry for that and have fixed it for the revised version.*

page 30, figure 6: I find it slightly confusing to see "transparent cirrus" with IOT up to (and possibly exceeding) 4. Perhaps a different word would be more suitable (but I don't feel too strongly about this).

*We understand the reviewer, but if possible we would like to stick to the word "transparent". We have clarified in Sect. 4.3.1 and Sect. 4.4.1 that the transparency in this context is related to the saturation of the CALIOP laser beam rather than to the normal sense of the term by adding the sentence: "Also note that the terms "transparent" and "opaque" in this context are solely related to the saturation of the CALIOP laser beam and tells whether it was able to fully penetrate the cirrus cloud (transparent cirrus) or not (opaque cirrus)".*

page 33, figure 9: There is some evidence that 2D-histograms with hexagonal binning are superior to those with rectangular binning; the authors may wish to search the internet for the keyword "hexplot", try to show the data or a hexagonal grid, and judge for themselves. An article illustrating why it may be superior can be found at http://www.meccanismocomplesso.org/en/hexagonal-binning/

*We thank the reviewer for the suggestion and information about hexagonal binning. It was an interesting article and it seems that hexagonal binning can be superior in many cases. This is something we will have in mind for future visualisations, but for this manuscript we think that the squared binning is sufficient to show the general patters of the CiPS retrieval errors (MAPE and MPE).*